# Timing of spring events changes under modelled future climate scenarios in a mesotrophic lake

Jorrit P. Mesman[1], Inmaculada C. Jiménez-Navarro[2], Ana I. Ayala[1], Javier Senent-Aparicio[2], Dennis Trolle[3], Don Pierson[1]

[1]Department of Ecology and Genetics, Uppsala University, Uppsala, 75236, Sweden

[2]Department of Civil Engineering, Catholic University of San Antonio, Guadalupe, 30107, Spain

[3]WaterITech, Døjsøvej 1, Skanderborg, 8660, Denmark

*Correspondence to*: Jorrit P. Mesman (jorrit.mesman@ebc.uu.se)

*Keywords*: Phenology, Climate change, Ice, Stratification, Phytoplankton, Runoff, Process-based modelling

*Correspondence emails*: jorrit.mesman@ebc.uu.se; icjimenez@ucam.edu; isabel.ayala.zamora@ebc.uu.se; jsenent@ucam.edu; dt@wateritech.com; don.pierson@ebc.uu.se

**Abstract.** Lakes experience shifts in the timing of physical and biogeochemical events as a result of climate warming, and especially relative changes in the timing of events may have important ecological consequences. Spring in particular is a period in which many key processes that regulate the ecology and biogeochemistry of lakes occur, and also a time which may experience significant changes under influence of global warming. In this study, we used a coupled catchment-lake model forced by future climate projections to evaluate changes in the timing of spring discharge, ice-off, the spring phytoplankton peak, and the onset of stratification, in a mesotrophic, temperate lake. Although the model explained only part of the variation in these events, the overall patterns were simulated with little bias. All four events showed a clear trend towards earlier occurrence with climate warming, with ice cover tending to disappear at the end of the century in the most extreme climate scenario. Moreover, relative shifts in the timing of these springtime events also occurred, with the onset of stratification tending to advance slower than the other events, and the spring phytoplankton peak and ice-off advancing faster in the most extreme climate scenario. The outcomes of this study stress the impact of climate change on the phenology of events in lakes and especially the relative shifts in timing during spring. This can have profound effects on food-web dynamics as well as other regulatory processes, and influence the lake for the remainder of the growing season.

## 1. Introduction

A changing timing of lake physical and biogeochemical events is one of the many consequences of climate change. Long-term changes in event timing have been reported for, for instance, the onset and end of stratification (Woolway et al., 2021; Moras et al., 2019), the onset and end of ice cover (Sharma et al., 2019), lake metabolism (Ladwig et al., 2022), the spring phytoplankton bloom (Peeters et al., 2007a; Gronchi et al., 2021; Meis et al., 2009), the spring zooplankton peak (Straile, 2000; Anneville et al., 2002) and fish spawning (Jeppesen et al., 2012; Lyons et al., 2015). Moreover, trends in stratification, ice cover, and plankton phenology are likely to continue under future climate warming (Woolway et al., 2021; Feldbauer et al., 2022; Gronchi et al., 2023). Such shifts in timing are highly relevant for lake ecosystem functioning, as they may lead to an altered duration of the growing season (Rouse et al., 1997), changed biogeochemical conditions during key biological events (Weyhenmeyer et al., 2013; Prowse and Brown, 2010; Adrian et al., 2012), or a trophic mismatch in cases where the relative timing of multiple processes changes (Donnelly et al., 2011; but see Berger et al., 2014; Thackeray et al., 2010). Events during critical time windows, and the antecedent lake conditions during these periods, are highly relevant throughout the year, and effects may persist beyond the event itself (Adrian et al., 2012). For instance, antecedent lake conditions preceding storms may be more important than storm characteristics themselves to determine storm effects (Thayne et al., 2021), and autumn phytoplankton blooms may or may not trigger depending on mixing conditions during turnover (Findlay et al., 2006), which may again affect phytoplankton composition in the following spring (Yang et al., 2016a). Incomplete winter mixing, due to warm winter temperatures or mild winds, affects oxygen conditions in following years (Schwefel et al., 2016). It is especially in spring, however, that many key events for the food web occur (Adrian et al., 2012; Sommer et al., 2012) that may resonate for the remainder of the season (Straile, 2005), and observations and simulations suggest that, in general, there will be an earlier occurrence of springtime events in mid- to high-latitudes lakes with climate warming (e.g. Winder and Schindler, 2004; Woolway et al., 2021; Feldbauer et al., 2022). These changes can influence ecosystem functioning for the remainder of the growing season and thus represent a latent consequence of climate warming.

While there are several key events regarding ecosystem functioning at play during spring, previous studies have typically focused only on one or few of these. The coupled phenology of phytoplankton and zooplankton is rather well-studied (Sommer et al., 2012), but timing of other events is often studied in isolation, or restricted to seasonality in lake physical processes (ice cover and stratification). Hence, we do not know if the changes in the timing of these spring events are changing synchronically as a consequence of climate change. We investigated

and compared changes of several key events during spring, namely the timing of ice-off, spring discharge, the
spring phytoplankton bloom, and onset of stratification, in a mesotrophic lake in Sweden. Ice-off is relevant for
instance for its role in water column light availability, and a renewed exchange between water and atmosphere in
general. Spring discharge can be an important source of external nutrients in catchments with significant snow or
ice components. The spring phytoplankton bloom marks the start of the growing season, provides food for higher
trophic levels, and influences nutrient and oxygen concentrations. The onset of stratification is a key event as well,
and controls distribution of substances in the water column and affects, amongst others, oxygen, nutrient, and
phytoplankton dynamics until stratification breakdown in autumn. These events are all influenced by
meteorological conditions, but they also influence each other. Break up of snow-covered or white ice strongly
increases light penetration into the water (Weyhenmeyer et al., 2022), which is important both for spring
phytoplankton growth and formation of thermal stratification. In catchments with snow cover, spring high flows
may provide an important source of nutrients for the phytoplankton community (Hrycik et al., 2021). Lastly,
following turbulent water conditions in deep lakes, the onset of stratification is often a prerequisite for the spring
phytoplankton bloom (Huisman et al., 1999; Peeters et al., 2007b). Despite this, these events are rarely studied
together in a single lake, and the separate projected trends in timing are seldom compared to each other within the
same study site.

An earlier occurrence of spring events has several major consequences for lakes, but relative shifts could also
result in previously unforeseen ecosystem effects, as biogeochemical cycles can shift and ecological niches may
close or open due to changing time windows. For example, the timing of (spring) discharge in relation to the onset
of stratification may partially determine where external nutrients end up in the water column (Fink et al., 2016;
Cortés et al., 2017) and therefore their fate during the growing season. A longer gap between ice-off and the onset
of stratification would alter mixing conditions early in the year, with corresponding changes in phytoplankton
composition (Winder and Sommer, 2012). If phytoplankton growth is reliant on inflow of external nutrients, a
shift towards earlier inflow, to periods with unfavourable light conditions for growth, might affect the intensity of
a spring phytoplankton bloom (Hrycik et al., 2021), with consequences for higher trophic levels as well.

We used a coupled catchment-lake model framework to make future projections of the timing of these four events
(ice-off, spring discharge, the spring phytoplankton bloom, and onset of stratification) and additionally to compare
the projected trends between each of them. The use of process-based models can provide a robust framework for
future projections of the timing of these springtime events, and the numerical coupling of lakes to their catchment
allows a more thorough evaluation of climate change impacts and environmental changes (Kong et al., 2022). We

hypothesised that all events would occur earlier in the year in a future, warmer climate, which is in line with previous studies, but also that relative changes in the timing of these events would occur. The latter expectation was partially due to the different processes driving each event, for example early-spring rain and air temperature would have the greatest importance in affecting snow and ice melt, while wind and temperature later in the season would affect the onset of stratification. Moreover, the effect of the strong seasonal cycle of solar radiation at the latitude of our study site would provide different physical constraints on phytoplankton, stratification, ice-off, and discharge. Climate warming could therefore affect not only the timing of these events, but also how they depend on each other and other external forcing – for example, in a future climate, the spring phytoplankton bloom might no longer rely on ice-off, but on the seasonal increase in solar radiation. The aim of our study is to create future projections of the timing of ice-off, spring discharge, the spring phytoplankton bloom, and onset of stratification and assess their absolute and relative changes, in order to better understand the impact of climate change on springtime events in lakes.

## 2.    Material & Methods

### 2.1. Study site

Lake Erken is located in eastern Sweden (N 59.8°, E 18.6°) and has a mean depth of 9 m, a maximum depth of 21 m, and covers 24 $km^2$ (Fig. 1). It is considered mesotrophic, with a summer average Secchi depth of 4.2 m, a surface total phosphorus concentration of 21.9 $mg/m^3$, and a surface chlorophyll concentration of 5.6 $mg/m^3$. The catchment of the lake has a maximum elevation difference of around 50 m and is covered mostly by pine forest, interspersed by deciduous forest and farmland. Around 50% of the catchment is drained by a stream that enters the lake at its western end (Fig. 1) and the hydraulic retention time is around 7 years. Weather data were collected on an island in the lake and missing data were supplemented with nearby weather stations (Moras et al., 2019). Lake data were collected near the deepest point of the lake (Fig. 1) and all data are publicly available at the Sites Data Portal (2022).

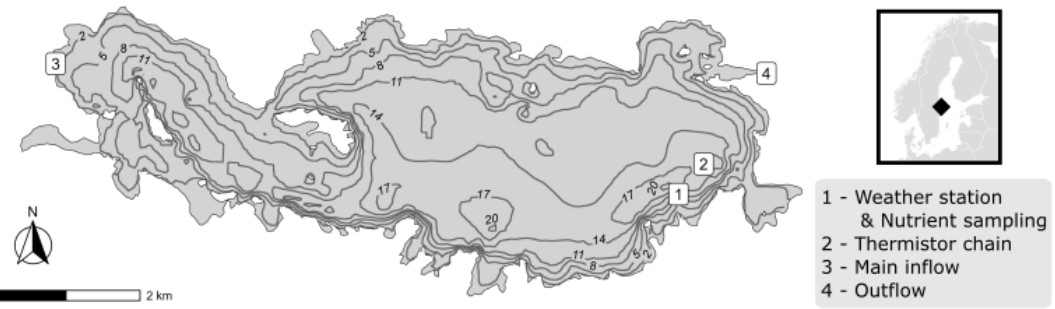

**Figure 1. Bathymetric map of Lake Erken and the locations where data were collected.**

## 2.2. Model framework and model performance

The present study builds upon a coupled catchment-lake model setup created by Jiménez-Navarro et al. (2023). This model setup was used to simulate catchment discharges, nutrient loads, and in-lake conditions under present and future conditions, and in the present study, we additionally assessed simulations of spring events. SWAT+ is a catchment model that takes into account meteorological forcing and catchment characteristics, such as land use and soil type, to reproduce catchment hydrology (Bieger et al., 2017) and was used to simulate discharges into

Lake Erken. Stream nutrient concentrations and temperatures were estimated statistically using LOADEST (Runkel et al., 2004; Runkel and De Cicco, 2017) and air2stream (8-parameter version, Toffolon and Piccolroaz, 2015; Piccolroaz et al., 2018), respectively. The coupled GOTM-WET model was used to simulate lake physics and biogeochemistry. GOTM simulates one-dimensional lake physics based on meteorological and hydrological boundary conditions (Umlauf et al., 2005) and WET is a modular biogeochemical model that can simulate amongst others nutrient, phytoplankton, zooplankton, and fish dynamics (Schnedler-Meyer et al., 2022). Light

absorption by components in the WET model (inorganic matter, particulate organic matter, and phytoplankton biomass) feeds back to the physical model. The simulated food web composition for this study involved four phytoplankton groups (diatoms, cyanobacteria, green algae, and flagellates), one macrophyte group, and one zooplankton group. The models were calibrated using data collected locally as part of the Lake Erken monitoring

program using the time period 2000-2015 (2007-2015 for inflow data), and the period 2016-2021 was used for model validation. The calibrated models were then run under future climate projections, where SWAT+ output was used as input into GOTM-WET. These future climate projections were based on five GCMs (General Circulation Models, models used: BCC-CSM2-MR, CanESM5, INM-CM5-0, MiroC6, MRI-ESM2-0) from CMIP6 (Coupled Model Intercomparison Project, Eyring et al., 2016) and ran from 1985 until the end of the 21$^{st}$

century. Each GCM provided the meteorological forcing required to run both SWAT+ and GOTM-WET and the projections were bias-corrected to locally observed meteorological data using quantile mapping (see Jiménez-

Navarro et al., 2023). Two socioeconomic pathways, SSP 2-45 and SSP 5-85 were used, corresponding to a future with moderate or no climate mitigation efforts, respectively. The period 1985-2014 was the same for both scenarios (historical period) and the two pathways diverged over the period 2015-2100. A full description of the different models, the coupling of the models and the employed calibration techniques, can be found in Jiménez-Navarro et al. (2023).

A comparison between simulated and observed inflow and lake data, spanning 2000-2021 for most variables, confirmed that the models reproduced the dynamics of the system with reasonable accuracy (see Jiménez-Navarro et al., 2023). Discharge, lake temperature and oxygen concentrations were simulated well, with $R^2$ values over 0.6, whereas nutrient and chlorophyll were more uncertain ($R^2$ values between 0.1 and 0.6 for $NH_4$, $NO_3$, $PO_4$, and chlorophyll), although the model still reproduced convincing seasonal cycles (for a more detailed assessment of model performance, see Supplement section S1 and Jiménez-Navarro et al. (2023)). This model performance for biogeochemical variables was in a similar range as previous studies (e.g. Chen et al., 2020; Kong et al., 2022; Zhan et al., 2023). As such, we took the calibrated SWAT+-GOTM-WET model framework as an acceptable representation of the ecosystem and used it as a basis to look at springtime phenology. In the Results section, we provided a separate assessment of the model's reproduction of the springtime phenology and in the Supplementary Material (Section S2), we provide information about additional model validation, following the framework of Hipsey et al. (2020).

**2.3. Springtime events and other lake variables**

Four different springtime events were considered in this study: ice-off, date of 50% cumulative spring discharge, the spring phytoplankton bloom, and stratification onset.

Ice-off dates in the lake were recorded when the majority of the lake had thawed (earliest in 2000-2022 record: February 8; latest: April 26; median: April 4). The GOTM-WET model contained an ice module, but because snow was not considered, ice-on dates were typically accurate (mean absolute error 10 days, mean error -4 days), but ice-off dates were simulated consistently too early (mean absolute error 22 days, mean error -22 days, using the GOTM ice module). We therefore instead used a threshold of surface water temperature to decide the day of modelled ice-off. Multiple thresholds were tested with intervals of 0.5 °C and we settled on 2 °C, which showed the lowest bias (mean absolute error 12 days, mean error 2 days). The first day the modelled surface water temperature passed this threshold was set to be the day of ice-off. The date of ice-off was set to the day of the year with lowest surface water temperature in case no ice was simulated, which was necessary to account for ice-free years under future climate simulations.

The date of 50% cumulative spring discharge was chosen as indicator of the timing of spring snowmelt runoff. We followed an identical approach to Hrycik et al. (2021), where discharge was summed between January 1 and May 31, and the day that the cumulative runoff passed 50% of the total was calculated.

A peak of chlorophyll was used as indication for the spring phytoplankton bloom. In most years, a single spring peak in chlorophyll was visible in the observed data in spring, but in several years, there were similar, separate peaks, necessitating a different approach than simply choosing the date of the highest peak, as we wanted to assess the timing of the first spring peak. Instead, we first determined the highest chlorophyll peak in the period January-May, and in the case of multiple peaks we then took the first peak that had at least 90% of the chlorophyll of the

highest peak. Although we applied this method to both the simulated and observed data, it should be noted that observed chlorophyll data were available at roughly weekly intervals during the spring period. Therefore, there was an uncertainty in the timing of the observed peak of about 1 week, in addition to the possibility of missing a short-lived bloom.

Onset of summer thermal stratification was taken as the day that a density difference between the surface and

180 bottom of more than 0.1 kg/m$^3$ formed (Wilson et al., 2020), at a surface water temperature above 4 °C, for at least seven consecutive days.

In addition to these springtime events, several other variables were calculated that could shed light on the reason behind the simulated trends. These were the chlorophyll concentration during the spring peak, the cumulative discharge in spring (January-May), the average ice thickness during ice cover, the average strength of stratification

(Schmidt stability, Schmidt, 1928; calculated with the R package "rLakeAnalyzer", Winslow et al., 2019) and mixed layer depth during the stratified period (using a density difference of 0.1 kg/m$^3$ from the surface, following Wilson et al., 2020). Moreover, the end of stratification was calculated in the same way as its onset, and onset of ice cover was based on the GOTM ice module. Finally, the total number of stratified or ice-covered days per year was evaluated as well.

**2.4. Trend estimation**

The timing of the springtime events and the other variables were calculated for each year in the climate scenarios and determined for each GCM separately. Following this, the results from the GCMs were averaged and a Mann-Kendall test was done to estimate trends over time, expressed as Sen's slope, using the R package "modifiedmk" (Patakamuri and O'brien, 2021). An intercept was estimated in addition to the Sen's slope, following Helsel et al.

(2020); this intercept refers to the value at the start of the simulation in 1985. For the cross-comparison of the

timing of springtime events, Mann-Kendall tests were additionally done for the trends in timing relative to other springtime events (e.g. the number of days the spring chlorophyll peak occurred before the onset of stratification). All analyses were done in R version 4.1.3 (R Core Team, 2022) and forcing files, scripts, and model setups are provided by Mesman et al. (2024).

## 3. Results

### 3.1. Model performance

In most years, the timing of the spring events was simulated closely to observations (70% of the events within 10 days of observed) and showed little bias (mean error < 5 days) (Fig. 2). However, only 29 - 47% of the variation was explained by the model, which was largely due to several years showing large discrepancies between observed and modelled results (Fig. 2). We investigated all events that were missed by more than 14 days, to discern whether this would invalidate the use of our model under future conditions (see Supplement section S3). Upon this further inspection, we concluded that for the five badly simulated years for discharge and chlorophyll, the model did indeed not capture the dynamics of the lake or catchment, though without indication that particular events led to a systematic over- or underestimation. However, for ice-off, poor fits were rather caused by the method of determining the date of ice-off, as the well-simulated surface water temperature (see Supplement section S1) was not always a useful predictor of the date of ice-off. Particularly, this occurred in years with short ice cover duration, in which the 2 °C threshold may estimate ice-off to occur too late. Similarly, for onset of stratification, sometimes a temporary period of stratification was identified as onset in the simulation, whereas in the observations a following period was taken as onset, despite bottom-top density difference being simulated accurately by the model. As such, we concluded that it was noise in water temperature observations that caused the threshold method to occasionally fail, rather than an inability of the model to simulate the state of the lake. Since the simulation provided good results in the majority of the years, the metrics only occasionally gave false impressions, and method failures would not strongly bias future predictions, we concluded that the method would overall give reliable estimates under future climate scenarios. Moreover, the lack of bias indicates that the model can provide the average timing of spring events under prevailing atmospheric conditions, even though year-to-year variability may be missed.

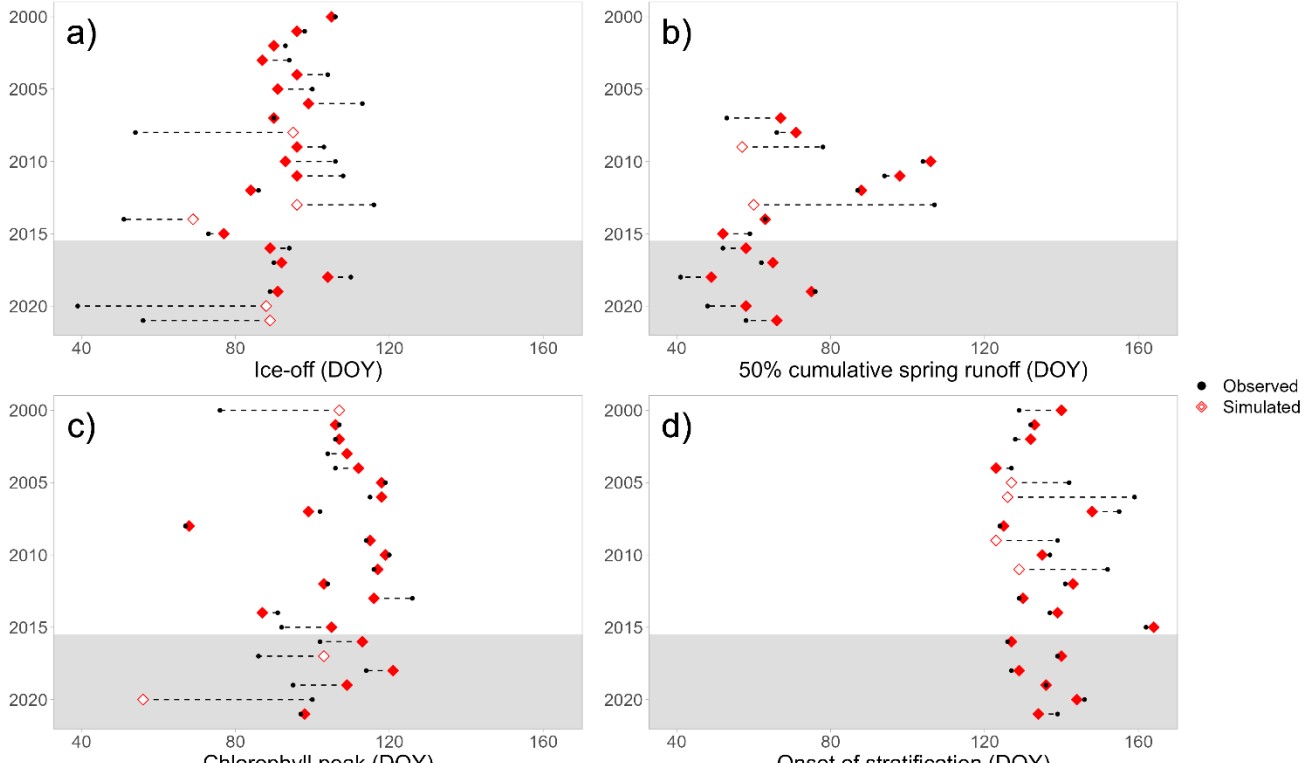

**Figure 2. Simulated (red diamonds) and observed (black circles) timing of (a) ice-off, (b) 50% cumulative spring runoff, (c) spring chlorophyll peak, and (d) onset of stratification. The years are on the y-axis, and the difference in timing is shown by a dashed line. The units on the x-axis are in day-of-year (DOY). The light grey area indicates the validation period. Open diamonds denote the years that were fitted badly (> 14 days error) and that are further investigated in Supplement section S3.**

### 3.2. Trends over time under climate scenarios

The duration of stratification increased (1.95 and 3.27 more stratified days per decade for SSP 2-45 and 5-85, respectively, Table 1), as did the strength of stratification, expressed as Schmidt stability (6.35 and 10.49 $J/m^2$/decade, Table 1). The mixed layer depth showed a tendency to become slightly shallower (-0.02 and -0.06 m/decade, Table 1). The magnitude of the spring chlorophyll peak decreased, by 0.37 (SSP 2-45) and 0.35 (SSP 5-85) $mg/m^3$/decade, while the cumulative spring discharge increased ($3.05 \cdot 10^5$ and $4.65 \cdot 10^5$ $m^3$/decade for SSPs 2-45 and 5-85, Table 1). Winter conditions became less severe, as the number of days with ice cover decreased (-5.68 and -7.02 days/decade, Table 1) and average ice thickness decreased by 0.012 and 0.014 m/decade. The percentage of ice-free winters increased from 3% in the first 30 years of the simulation to 38% under SSP 2-45 or 70% under SSP 5-85 at the end of the century, though the results varied strongly between the different GCMs (between 43% and 97% for the SSP 5-85 scenario).

**Table 1. Results of Mann-Kendall trend tests for trends during the future climate scenarios. DOY stands for day-of-the-year. Average values for the periods 1985-2014, 2040-2069, and 2070-2099 can be found in Supplement section S4.**

| Variable | Unit | SSP 2-45 | | | SSP 5-85 | | |
|---|---|---|---|---|---|---|---|
| | | p-value | Sen's slope (/decade) | Intercept | p-value | Sen's slope (/decade) | Intercept |
| Chlorophyll peak date | DOY | <0.0001 | -2.45 | 109.22 | <0.0001 | -3.71 | 112.11 |
| Peak spring chlorophyll concentration | mg/m$^3$ | <0.0001 | -0.37 | 14.42 | <0.0001 | -0.35 | 14.06 |
| 50% spring discharge date | DOY | <0.0001 | -2.21 | 77.61 | <0.0001 | -2.26 | 73.31 |
| Cumulative spring discharge | m$^3$ | <0.0001 | $3.05 \cdot 10^5$ | $8.69 \cdot 10^6$ | <0.0001 | $4.65 \cdot 10^5$ | $8.39 \cdot 10^6$ |
| Ice-off date | DOY | <0.0001 | -2.13 | 105.93 | <0.0001 | -3.75 | 107.95 |
| Ice-on date | DOY | <0.0001 | 3.75 | 2.05 | <0.0001 | 4.73 | 1.74 |
| Number of days with ice | days | <0.0001 | -5.68 | 73.34 | <0.0001 | -7.02 | 67.91 |
| Average ice thickness | m | <0.0001 | -0.012 | 0.14 | <0.0001 | -0.014 | 0.13 |
| Stratification onset | DOY | <0.0001 | -1.20 | 140.94 | <0.0001 | -1.88 | 141.68 |
| End of stratification | DOY | <0.0001 | 0.74 | 261.73 | <0.0001 | 1.42 | 260.74 |
| Number of stratified days | days | <0.0001 | 1.95 | 121.87 | <0.0001 | 3.27 | 118.02 |
| Average Schmidt stability during stratification | J/m$^2$ | <0.0001 | 6.35 | 175.19 | <0.0001 | 10.49 | 163.07 |
| Average mixed layer depth during stratification | m | 0.023 | -0.02 | 6.47 | <0.0001 | -0.06 | 6.50 |

### 3.3. Spring events timing under climate scenarios

The investigated spring events were without exception projected to occur earlier in the year, with stronger changes predicted for SSP 5-85 compared to SSP 2-45 (Fig. 3, Supplement section S4). Although there was substantial variation between the different GCMs, the negative Sen's slope was significant for all variables and climate scenarios (Table 1). However, the magnitude of the projected slope was different between the investigated variables, with for example the trend in the timing of the chlorophyll peak advancing roughly twice as fast as the trend in onset of stratification. A cross-comparison of the relative trends revealed that some timings of spring events indeed significantly changed relative to each other (Fig. 4). More specifically, the rate of change of the onset of stratification was slower than that of other events, while for the SSP 5-85 climate scenario, the advance of the spring chlorophyll peak and ice-off were faster than that of the other two events.

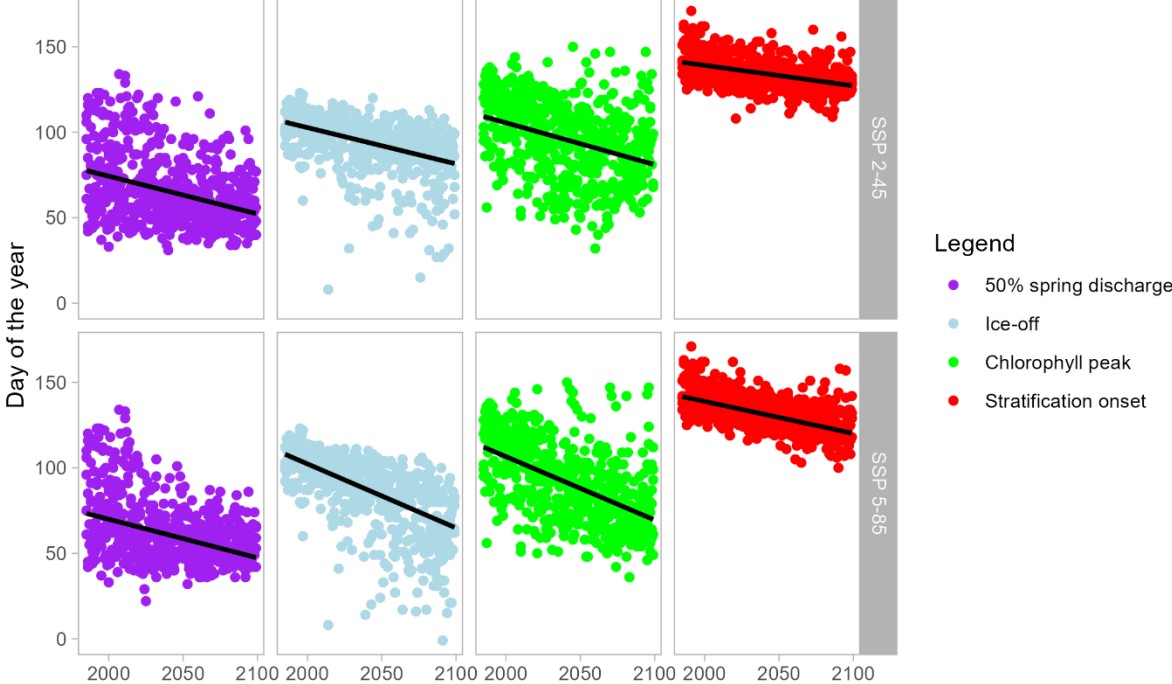

**Figure 3. Projected timings of spring events for 1985-2100, with the upper row showing the projections for SSP 2-45 and the lower row for SSP 5-85. Results for all GCMs are plotted here. The black line indicates the fit of the Mann-Kendall trend test on the ensemble mean (details of the Mann-Kendall test results can be found in Table 1).**

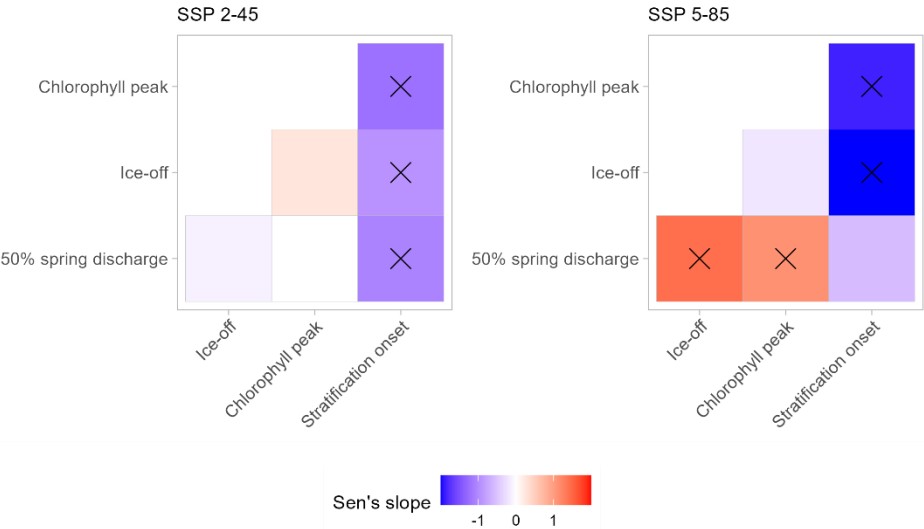

**Figure 4. Sen's slopes (days/decade) of timings of spring events relative to the other spring events. The colour scale indicates Sen's slope of the day-of-the-year of the event on the y-axis relative to the timing of the corresponding event on the x-axis. A positive slope therefore means that the variable on the x-axis advanced to earlier dates faster than the variable on the y-axis. For example, under SSP 5-85, ice-off date advanced faster than the date of the 50% discharge. Crosses indicate a significant difference from 0 (Mann-Kendall test, p-value < 0.05). For the exact values of the slopes and the p-values, see Supplement section S5.**

## 4.      Discussion

Each of the investigated spring events was projected by the model simulations to occur earlier in the year. As air temperatures in our climate scenarios were rising (Jiménez-Navarro et al., 2023), the earlier occurrence of ice-off and stratification onset was unsurprising and the rates of change were indeed in line with previous studies (Woolway et al., 2021; Magee and Wu, 2017; Shatwell et al., 2019; Feldbauer et al., 2022; Li et al., 2022). Under SSP 5-85, ice cover is projected to largely disappear in Lake Erken at the end of the century. Together with the shifting precipitation and runoff patterns, this will mean a complete transformation of the lake's winter conditions, with both societal and ecological relevance (Cavaliere et al., 2021; Knoll et al., 2019). The method to estimate ice-off from the model results (a 2 °C threshold) tended to simulate ice-off too late in years with low ice cover. Therefore, our study is likely underestimating the advancement rate of ice-off date, and ice may be disappearing even faster than the rates predicted here. Regarding the earlier discharge, the Lake Erken catchment is commonly snow-covered in winter, and future increasing air temperatures lead to earlier snowmelt and a concurrent discharge peak, a process common at high latitudes (Hrycik et al., 2021). Moreover, the future climate scenarios suggested that an increase in precipitation will occur during winter for the location of Lake Erken (Jiménez-Navarro et al., 2023), but projections of such precipitation patterns in future climate vary geographically (e.g. IPCC, 2021). As such, the earlier occurrence of spring discharge should be viewed as a phenomenon linked to areas where the accumulation of snow has an important effect on the regional hydrology and where winter precipitation is predicted to increase. Lastly, the earlier occurrence of the spring chlorophyll peak with climate change could be due to a combination of factors. Earlier ice-off promotes an earlier onset of phytoplankton growth, if light obstruction by ice is the main limiting factor for growth (Gronchi et al., 2021). Earlier spring discharge into Lake Erken also provides an earlier supply of nutrients, but in this lake, the majority of spring discharge occurs prior to ice-off and the spring chlorophyll peak. In Lake Erken, spring phytoplankton growth is not reliant on stratification due to the limited mean depth of the lake, and the spring chlorophyll peak (dominated by diatoms) tends to occur prior to onset of stratification (Figure 2; Weyhenmeyer et al., 1999; Moras et al., 2019), - an order of events which is commonly reversed in deeper lakes where stratification is required to overcome light limitation (Huisman et al., 1999; Gronchi et al., 2021). Altogether, the earlier chlorophyll peak therefore seems to be mostly attributable to the increased availability of light due to earlier ice-off, causing growth to commence earlier. The peak of the spring chlorophyll bloom (i.e. the end of net growth) in the model seemed to have been dictated mostly by nutrient limitation, as nutrient concentrations reached low concentrations around the time of the simulated peak, with a potential shift to more light limitation at the end of the century (see Supplement section S6). The role of

zooplankton grazing was predicted to increase under future climate projections as well, although a clear link between simulated the spring chlorophyll peak and zooplankton concentrations was not observed (see Supplement section S7), leading to nutrient and light limitation as the major determinants of the spring chlorophyll peak in the model.

These absolute changes in the timing of spring events may lead to several changes in lake state (Figure 5). For instance, the earlier spring chlorophyll peak leads to an earlier uptake of nutrients and a longer growing season. However, not all effects are restricted to spring itself. An earlier onset of stratification leads to lower hypolimnetic oxygen concentrations in the summer (Jane et al., 2023) and a longer period of nutrient limitation in the epilimnion (Sommer et al., 2012). These trends were indeed seen in our future projections, with chlorophyll and epilimnetic nitrate concentrations remaining constant despite increased nutrient loading (Jiménez-Navarro et al., 2023), and were likely partially driven by the earlier stratification onset.

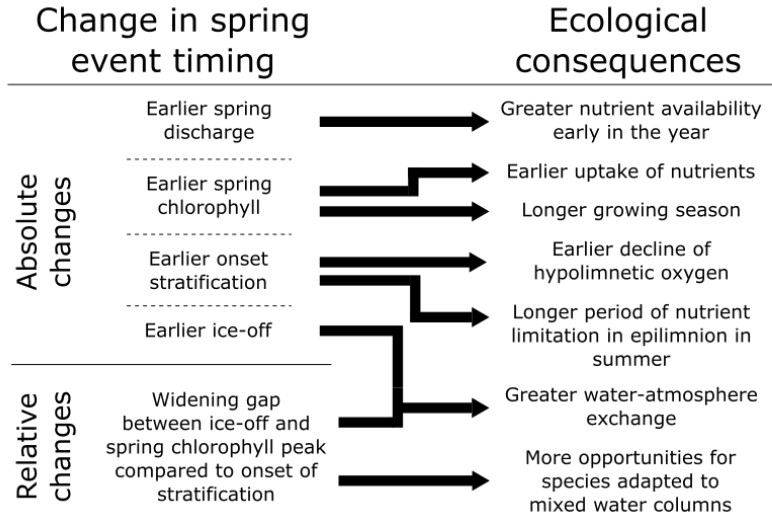

**Figure 5. Conceptual diagram of how the changes in spring event timing in Lake Erken under a warming climate, as simulated by the GOTM-WET and SWAT+ models, would link to ecological consequences in the lake.**

Although the predicted changes in event timing are reported as linear trends, it should be noted that we do not assume that these changes are entirely linear. Especially in SSP 2-45, the development of air temperature through the simulation period is not linear, and the timing of events will not follow a linear trend over time either. In Supplement section S4, averages in separate time periods are reported, and for instance the advance in timing of the spring chlorophyll peak gives an indication of slowing down or stopping in the second half of the century under SSP 2-45. Reported linear changes should therefore be seen as the average change over the period 1985-2100, and we did not investigate the shape of the trend during this period.

The absolute changes in the timing of spring events were comparable to findings in other studies, but relative changes (other than phytoplankton-zooplankton dynamics) have received much less attention in the scientific

literature, despite their potential impact on lake ecosystem functioning. One of the few studies to look at relative changes, Meis et al. (2009) found no effects of timings of ice-off and stratification on phytoplankton spring phenology, but rather a secondary effect of temperature on the dominant phytoplankton species. Earlier onset of stratification is a well-known consequence of climate warming (Woolway et al., 2021), but our findings suggest

that other events in spring will advance at an even higher rate in our study lake. This leads to an increased gap between onset of stratification and the three other events evaluated here. Such a differential effect on lake events can lead to marked changes in lake dynamics, potentially affecting food web dynamics (e.g. Thackeray et al., 2008; Yang et al., 2016b). In the SSP 5-85 scenario, the rate of an earlier spring chlorophyll peak and an earlier ice-off exceeded that of the two other spring events. The similar trend of ice-off and the spring chlorophyll peak

is in line with findings by Gronchi et al. (2021), who postulated that the onset of the spring bloom in lakes like Lake Erken (i.e. light-limited, but phytoplankton growth not reliant on stratification) is either dependent on ice-off or the seasonal increase in solar radiation. The former option would suggest that the trend in the onset of the spring bloom follows that of ice-off, whereas the latter option implies only a weak response to climate warming. Although Gronchi et al. (2021) looked at the onset of the growth, whereas we looked at the peak of the spring

phytoplankton bloom, the similar trend in our study would confirm ice-off as main determinant for the timing of the spring chlorophyll peak. The timing of the spring bloom in Lake Erken indeed tended to occur around or shortly after the time of ice-off (Fig. 3; Weyhenmeyer et al., 1999). The earlier spring chlorophyll peak under warmer climate projections coincided with a lower chlorophyll concentration during the spring peak. The earlier occurrence of ice-off would move the start of phytoplankton growth to days with less incoming solar radiation, a

strong effect due to Lake Erken's high latitude, and shorter, less intense winters may cause higher zooplankton concentrations at ice-off (Hebert et al., 2021), which could both partially explain the less intense spring peaks. The climate change scenarios reveal pertinent changes in the conditions of Lake Erken towards the end of the century, including longer and stronger stratification, shorter ice cover, and absolute and relative changes in the timing of springtime events. The longer period between ice-off and the spring bloom on one hand and the onset

of stratification on the other, could open up new niches for species adapted to well-mixed water columns, such as diatoms (Fig. 5; Yang et al., 2016b). Regarding summer phytoplankton dynamics, the earlier onset of stratification lengthens the period of nutrient limitation in the epilimnion and this may partially explain the lower summer chlorophyll concentrations at the end of the century, despite higher yearly average nutrient values (Jiménez-Navarro et al., 2023; Figure 5). Indeed, previous studies have shown that climate warming and a shifting timing

of spring events may alter food web composition (Beare and McKenzie, 1999; Winder and Schindler, 2004;

Thackeray et al., 2008), and that events in winter and spring can have effects well into the following summer and beyond (Straile, 2005; Hampton et al., 2017).

Methods to make future projections of ecological conditions are by definition uncertain, and the present study is no different. For example, the method of determining the timing of events was sometimes not in line with the observed data, where real patterns are often more complex than a single peak or event. Zooplankton grazing can be an important factor for spring phytoplankton (e.g. Peeters et al., 2007b), but the simulated zooplankton could not be validated due to a lack of long-term, high-frequency zooplankton data. Moreover, scenarios of future nutrient loads were done in a simple approach (Jiménez-Navarro et al., 2023), which did not consider, for example, potential changes in land use policies. Regardless, the models showed clear signs of earlier spring events under warmer climate conditions, in line with previous studies, and the coupled setup of the catchment and lake allowed future projections that took into account the interdependency of the lake and its catchment. Changing phenology is an important aspect of ecosystems' response to climate change and this study provides more insight into relative temporal changes between different springtime events in lakes under future climate scenarios.

**Conclusions**

We analysed future climate projections of the timing of spring discharge, ice-off, spring chlorophyll peak, and onset of stratification in a mesotrophic lake. While all events occurred earlier in the year under a warmer climate, there were marked changes in the relative timing of events as well, with the onset of stratification advancing slower than the other events and both the spring chlorophyll peak and ice-off advancing faster under the most severe climate scenario. Phenological changes in individual lake events in response to climate change have been well-established, but relative changes and future projections of the timing of multiple interdependent events in the same lake, extending to biological and watershed responses, have received little attention so far. While changes in the timing of events have important consequences for ecosystems, relative changes may present a secondary, perhaps unforeseen, effect that can influence food web dynamics and lake functioning. The simulations in the present study imply that both absolute and relative changes in the timing of springtime lake events are likely to occur in response to climate warming, and that this should be considered when assessing climate change impacts on lakes.

**Code and Data availability**

Lake Erken meteorological and lake data are available at the SITES data portal (SITES Data Portal, 2022). The future climate scenarios were generated by the CMIP6 project (Eyring et al., 2016) and downloaded from the

DKRZ ESGF-CoG node.

A repository containing the files, model setups, and scripts that were used to produce the results of this study, has been made available by Mesman et al. (2024).

**Author contributions**

JPM and DCP designed the study. JPM and ICJN set up and ran the model simulations. JPM created the

visualisations and wrote the original draft. All authors were involved in discussions throughout the course of the study and in reviewing and editing the final manuscript.

**Competing interest statements**

The authors declare that they have no conflict of interest.

**Acknowledgements**

This study was funded by the European Union's Horizon 2020 research and innovation program, grant agreement number 101017861 (SMARTLAGOON). The study has been made possible by data provided by the Swedish Infrastructure for Ecosystem Science (SITES).

We acknowledge the World Climate Research Programme, which, through its Working Group on Coupled Modelling, coordinated and promoted CMIP6 and the Earth System Grid Federation (ESGF) for archiving the

data and providing access.

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
