# Peer review of "Timing of spring events changes under modelled future climate scenarios in a mesotrophic lake"

_EGUsphere, 2023_

## Author Comment (AC1)

General Comments

- Overall, this is a well-written and important contribution to our understanding of changing ecosystem functioning in lakes. It provides new insight by both developing projections of biogeochemical variables like phytoplankton dynamics and a novel comparison of the relative change in timing of important limnological events. The methods and results are clearly presented and the research is well contextualized. I suggest a few improvements below to better present the research in the context of other studies and research within the field of limnology.

> We thank the reviewer for the kind comments and for their useful suggestions, which allowed us to clarify and improve the manuscript. We respond to each comment below, with our responses in text boxes and proposed additions to the text in red. Added references are given at the end of our reply.

Specific Comments

- **1A** If you're not limited by words already, consider adding a sentence in the abstract that states how well your model did during training/validation to add support to the validity of your projections

> We will add the following sentence to the Abstract, related to the model performance:
>
> L. 18: Although the model explained only part of the variation in these events, the overall patterns were simulated with little bias.

- **1B** In the first few sentences of the intro, can you add some language to make it crystal clear whether the studies you are citing demonstrated *already observed* changes or projections in timing of processes? On a glance, I think most of the studies you cite are observed already and adding a short paragraph that more thoroughly summarizes findings from other projection studies would help highlight the novelty of your approach (including phytos and catchment loading and comparing relative changes in annual timing events across multiple variables)

> We will add a sentence with additional projection studies to link past observations to future projections:
>
> L. 31: Moreover, trends in stratification, ice cover, and plankton phenology are likely to continue under future climate warming (Woolway et al., 2021; Feldbauer et al., 2022; Gronchi et al., 2023).

- **1C** I think you are using spring 'metrics', 'events', 'processes' interchangeably to refer to your four response variables in the intro—might be good to choose one and stick with that

> This is a valid comment and in line with comment 2C of Reviewer 2. When referring to the four spring events (ice-off, onset stratification, etc.), we will now consistently use the term "events".

- **1D** You introduce some really good, but new, content in the last paragraph of the intro (line 59 on) about why relative differences in the timing of spring events matters. I wonder if you could make this its own paragraph before you introduce your study and hypotheses so that you can expand a bit more on why relative shifts in timing matter—this is the key finding from your study so it should be emphasized heavily in the intro

> We agree that it's beneficial to highlight the importance of relative shifts in timing more. We will add a new paragraph to the introduction and include some additional ways in which relative shifts could matter. Moreover, we will restructure the last paragraph in response to comments from Reviewer 2, so that the Introduction ends with aims and hypotheses, instead of new content.
>
> *Add after L. 51:* An earlier occurrence of spring events has several major consequences for lakes, but relative shifts could also result in previously unforeseen ecosystem effects, as biogeochemical cycles can shift and ecological niches may close or open due to changing time windows. For example, the timing of (spring) discharge in relation to the onset of stratification may partially determine where external nutrients end up in the water column (Fink et al., 2016; Cortés et al., 2017) and therefore their fate during the growing season. A longer gap between ice-off and the onset of stratification would alter mixing conditions early in the year, with corresponding changes in phytoplankton composition (Winder and Sommer, 2012). If phytoplankton growth is reliant on inflow of external nutrients, a shift towards earlier inflow, to periods with unfavourable light conditions for growth, might affect the intensity of a spring phytoplankton bloom (Hrycik et al., 2021), with consequences for higher trophic levels as well.

- **1E** You might want to add a citation in the introduction somewhere to Adrian et al. 2012 who discuss how changes in climate drivers during key time periods are critical to informing overall ecosystem function: Adrian, R., Gerten, D., Huber, V. *et al.* Windows of change: temporal scale of analysis is decisive to detect ecosystem responses to climate change. *Mar Biol* **159**, 2533–2542 (2012). https://doi.org/10.1007/s00227-012-1938-1

> This is indeed a relevant reference for the importance of critical events and their effects later in the season; thank you for this suggestion! We will now cite it to support our statements in the Introduction.

- **1F** Line 42: I suggest remove 'in this study' phrasing and focus on why these metrics are important generally in this paragraph before you emphasize the details of your study specifically

We will remove the phrase "In this study" and add several sentences to the Introduction that outline the importance of the four events under study.

*Add after L. 44:* Ice-off is relevant for instance for its role in water column light availability, and a renewed exchange between water and atmosphere in general. Spring discharge can be an important source of external nutrients in catchments with significant snow or ice components. The spring phytoplankton bloom marks the start of the growing season, provides food for higher trophic levels, and influences nutrient and oxygen concentrations. The onset of stratification is a key event as well, and controls distribution of substances in the water column and affects, amongst others, oxygen, nutrient, and phytoplankton dynamics until stratification breakdown in autumn.

- 1G Methods, line 97-102: can you provide reference to any other studies which use biogeochemical process models and have similar R2 for reproducing observations? I'm not implying that the fit isn't good enough, just that comparing to what others have done would be helpful to justify some of the lower R2 values

We will add references to three studies, which reported a similar goodness-of-fit for biogeochemical variables, also using coupled physical-biogeochemical models.

L. 102: This model performance for biogeochemical variables was in a similar range as previous studies (e.g. Chen et al., 2020; Kong et al., 2022; Zhan et al., 2023).

- 1H Line 108: can you provide a date range for the historical record of ice-off dates? Also in this section, can you report the bias for your 2C threshold for simulating ice-off for comparison since you report the error using the ice module?

A date range will be provided for ice-off dates and the MAE and ME will be also reported for the ice-off calculation with the temperature threshold.

New L. 108: Ice-off dates in the lake were recorded when the majority of the lake had thawed (earliest in 2000-2022 record: February 8; latest: April 26; median: April 4).

New L. 112: Multiple thresholds were tested with intervals of 0.5 °C and we settled on 2 °C, which showed the lowest bias (mean absolute error 12 days, mean error 2 days).

- 1I Line 145: include a citation for your workflow here as well?

Thank you for this suggestion: we will add the citation for the workflow.

New L. 145: All analyses were done in R version 4.1.3 (R Core Team, 2022) and forcing files, scripts, and model setups are provided by Mesman et al. (2023).

- **1J** Line 148-149: a sentence similar to this would add strength to the abstract in demonstrating that your model performed well against observations. I would suggest adding the years of this calibration/validation time period here (not necessary for abstract though I think)

> In line with comment 1A, we will add a sentence to the abstract.
>
> L. 18: Although the model explained only part of the variation in these events, the overall patterns were simulated with little bias.
>
> We decided to not split the results between calibration and validation period in the text. The percentage of events with an error less than 10 days was in fact the same for calibration and validation, but as can be seen in Figure 2, there were clear differences between the variables, so such a statement might give a wrong impression. Moreover, the number of years was necessarily limited and the validation period contained a small number of years, so we preferred to report the results for the whole period together. Figure 2 allows the reader to assess each variable and period separately.

- **1K** Results, line 170: maybe just me, but I'm not familiar with the term 'shoal'. Could you rephrase as 'increase' or 'decrease'?

> The term is indeed not used often; we will now use "become shallower" instead.

- **1L** Table 1: Is there a way you could visualize this rather than providing a table (but perhaps keep table in SI)? I'm envisioning something similar to Figure 2 where you show the difference between the value at the beginning of the simulation (intercept) and the mean value at the end of the projection time period based on Sen's slope? This would allow you to highlight the directionality and magnitude of average change

> We agree that a visualisation is often more intuitive than a table for readers, but we preferred to leave the table as is, because a) the table contains a large number of variables, and not all are the main focus of the paper, and b) each variable has different units, which would complicate a figure.

- **1M** Figure 2: can you make the font size overall a bit larger? It is necessary for me to zoom in quite a bit to read it as is. Would also suggest adding panel labels if this is a journal requirement. Instead of the purple square, maybe could you make the red diamonds open for years with a bad fit, filled for years with a good fit? The square is a little distracting (not a major issue though)

> We were indeed struggling with how to represent the badly-fitted years in the plot, and we really liked the idea of using open and closed diamonds for this. Thank you for this suggestion! We will additionally add panel labels.
>
> The small font size was an oversight, for which we apologise. This will be improved as well in the new Figure 2. The script in the workflow will be updated accordingly.

[Figure]

New Figure 2:

Caption Figure 2. Simulated (red diamonds) and observed (black circles) timing of (a) ice-off, (b) 50% cumulative spring runoff, (c) spring chlorophyll peak, and (d) onset of stratification. The years are on the y-axis, and the difference in timing is shown by a dashed line. The units on the x-axis are in day-of-year (DOY). The light grey area indicates the validation period. Open diamonds denote the years that were fitted badly (> 14 days error) and that are further investigated in Supplement section S2.

- 1N Line 215: is there a figure you can reference to support this? As it's written, it's unclear if you mean under current conditions or under future projections

Also in response to Reviewer 2's comment 2J, we will clarify the sentence and refer to Figure 2 and two references.

New L. 215: In Lake Erken, spring phytoplankton growth is not reliant on stratification due to the limited mean depth of the lake, and the spring chlorophyll peak (dominated by diatoms) tends to occur prior to onset of stratification (Figure 2; Weyhenmeyer et al., 1999; Moras et al., 2019), …

- 1O Line 245: this is really interesting. Did you calculate chlorophyll-a concentrations later in the growing season or just spring? I am wondering if there is an antecedent effect for later in the year which could have broader implications for additional bloom events and could be useful to add to the discussion

We focus in our study on the spring period, but have model output for the whole year. If the lower spring peak was indeed caused by a gradual shift to more light limitation compared to nutrient limitation (L. 220-221), while nutrient inputs stayed the same, one could expect a "broader" rather than "higher" spring peak, and this could have implications later in the season. It would be difficult to assess whether there would be causal links in our current model setup, however, and this would require additional experiments that would distract from the main message in the paper. However, also in response to Reviewer 2's comment 2A, we add a section to the Discussion on effects of spring events beyond spring itself.

*Add after L. 224:* These absolute changes in the timing of spring events may lead to several changes in lake state (Figure 5 *(an added diagram based on Reviewer 2's comment 2A)*). For instance, the earlier spring chlorophyll peak leads to an earlier uptake of nutrients and a longer growing season. However, not all effects are restricted to spring itself. An earlier onset of stratification leads to lower hypolimnetic oxygen concentrations in the summer (Jane et al., 2023) and a longer period of nutrient limitation in the epilimnion (Sommer et al., 2012). These trends were indeed seen in our future projections, with chlorophyll and epilimnetic nitrate concentrations remaining constant despite increased nutrient loading (Jiménez-Navarro et al., 2023), and were likely partially driven by the earlier stratification onset.

- **1P** Line 272: I think you should emphasize that this is especially true for biological responses like chla (there are studies looking at multiple connected hydrodynamic processes, Ayala et al. 2020, Barbosa et al. 2021, Feldbauer et al. 2022, Desgue-Itier et al. 2023, Wynne et al. 2023, etc.)

We agree, and will add a phrase that stresses the novelty of also looking into biological and watershed responses. The most notable exception of biological processes studied in conjunction would be phytoplankton-zooplankton phenology, which may not have been apparent in our Introduction, so we added a sentence about this as well.

L. 41: The coupled phenology of phytoplankton and zooplankton is rather well-studied (Sommer et al., 2012), but timing of other events is often studied in isolation, or restricted to seasonality in lake physical processes (ice cover and stratification).

New L. 270-272: Phenological changes in individual lake events in response to climate change have been well-established, but relative changes and future projections of the timing of multiple interdependent processes events in the same lake, extending to biological and watershed responses, have received little attention so far.

- **1Q** I think the study could benefit from more discussion of the implicit assumptions from focusing on spring event timing as your response variables (e.g., instead of summer, winter, or fall events). You do a good job justifying why spring is important (and I believe it), but I think you could add context which highlights other research which shows that antecedent conditions during other time periods (e.g., winter-time dynamics, storm events) are important for year-round functioning and adding some

context to acknowledge this would be helpful in the discussion. Some potential citations

- o Cavaliere et al. 2021 https://doi.org/10.1029/2020JG006165
- o Adrian et al. 2012 https://doi.org/10.1007/s00227-012-1938-1
- o Thayne et al. 2021: https://aslopubs.onlinelibrary.wiley.com/doi/full/10.1002/lno.11859

We add context regarding critical timing windows and antecedent conditions in other parts of the year, as suggested. We will mention storms, the occurrence of autumn blooms, and the effects of incomplete mixing in winter. We mention this in the Introduction, before going deeper into springtime events.

*Add after L. 35:* Events during critical time windows, and the antecedent lake conditions during these periods, are highly relevant throughout the year, and effects may persist beyond the event itself (Adrian et al., 2012). For instance, antecedent lake conditions preceding storms may be more important than storm characteristics themselves to determine storm effects (Thayne et al., 2021), and autumn phytoplankton blooms may or may not trigger depending on mixing conditions during turnover (Findlay et al., 2006), which may again affect phytoplankton composition in the following spring (Yang et al., 2016a). Incomplete winter mixing, due to warm winter temperatures or mild winds, affects oxygen conditions in following years (Schwefel et al., 2016).

Additional references, not previously cited in manuscript:

Adrian, R., Gerten, D., Huber, V., Wagner, C., and Schmidt, S. R.: Windows of change: temporal scale of analysis is decisive to detect ecosystem responses to climate change, Marine biology, 159, 2533-2542, 10.1007/s00227-012-1938-1, 2012.

Chen, W., Nielsen, A., Andersen, T. K., Hu, F., Chou, Q., Søndergaard, M., Jeppesen, E., and Trolle, D.: Modeling the Ecological Response of a Temporarily Summer-Stratified Lake to Extreme Heatwaves, Water, 12, 94, 10.3390/w12010094, 2020.

Cortés, A., MacIntyre, S., and Sadro, S.: Flowpath and retention of snowmelt in an ice-covered arctic lake, Limnology and Oceanography, 62, 2023-2044, 10.1002/lno.10549, 2017.

Findlay, H. S., Yool, A., Nodale, M., and Pitchford, J. W.: Modelling of autumn plankton bloom dynamics, Journal of Plankton Research, 28, 209-220, 10.1093/plankt/fbi114, 2006.

Fink, G., Wessels, M., and Wüest, A.: Flood frequency matters: Why climate change degrades deep-water quality of peri-alpine lakes, Journal of Hydrology, 540, 457-468, 10.1016/j.jhydrol.2016.06.023, 2016.

Gronchi, E., Straile, D., Diehl, S., Jöhnk, K. D., and Peeters, F.: Impact of climate warming on phenological asynchrony of plankton dynamics across Europe, Ecology Letters, 26, 717-728, 10.1111/ele.14190, 2023.

Jane, S. F., Mincer, J. L., Lau, M. P., Lewis, A. S. L., Stetler, J. T., and Rose, K. C.: Longer duration of seasonal stratification contributes to widespread increases in lake hypoxia and anoxia, Global Change Biology, 29, 1009-1023, 10.1111/gcb.16525, 2023.

Schwefel, R., Gaudard, A., Wüest, A., and Bouffard, D.: Effects of climate change on deepwater oxygen and winter mixing in a deep lake (Lake Geneva): Comparing observational findings and modeling, Water Resources Research, 52, 8811-8826, 10.1002/2016WR019194, 2016.

Thayne, M. W., Kraemer, B. M., Mesman, J. P., Ibelings, B. W., and Adrian, R.: Antecedent lake conditions shape resistance and resilience of a shallow lake ecosystem following extreme wind storms, Limnology and Oceanography, 67, S101-S120, 10.1002/lno.11859, 2021.

Winder, M. and Sommer, U.: Phytoplankton response to a changing climate, Hydrobiologia, 698, 5-16, 10.1007/s10750-012-1149-2, 2012.

Yang, Y., Stenger-Kovács, C., Padisák, J., and Pettersson, K.: Effects of winter severity on spring phytoplankton development in a temperate lake (Lake Erken, Sweden), Hydrobiologia, 780, 47-57, 10.1007/s10750-016-2777-8, 2016a.

Zhan, Q., de Senerpont Domis, L. N., Lürling, M., Marcé, R., Heuts, T. S., and Teurlincx, S.: Process-based modeling for ecosystem service provisioning: Non-linear responses to restoration efforts in a quarry lake under climate change, J Environ Manage, 348, 119163, 10.1016/j.jenvman.2023.119163, 2023.

---

## Author Comment (AC2)

The present study elaborated to investigate the future climate impacts on the spring hydrological and ecological processes (i.e. spring discharge, ice-off, spring phytoplankton peak, onset of stratification) in a typical temperate lake Erken. The findings have critical implications because these processes were rarely evaluated simultaneously and their different sensitivities to climate change may result in different change paces or rates, and eventually lead to profound consequences on lake ecosystem in the future. The paper is well-prepared and concisely written. I have a few major and minor concerns and would like to recommend publication if the authors can address them properly during the revision.

> We appreciate the compliments and also the concerns and suggestions raised by the reviewer. We respond to each comment below, with our responses in text boxes and proposed additions to the text in red. Added references are given at the end of our reply.

Major comments:

- 2A The manuscript repeatedly emphasizes the ecological consequences of the different rate of advancing among the four investigated events, but these are not actually evaluated and subject to inferences and speculation, which may be attributed to the limitation in the model. This can be compensated, and manuscript can be improved, if the author can add a conceptual diagram in the discussion section, which summarizes the findings from the study (i.e. different advancing rate among the processes, increasing gap between stratification onset and other processes, and the potential ecological consequences from literature, for example, increasing magnitude of winter diatom blooms, see e.g., Hebert et al., 2021 e2114840118 PNAS , or Kong et al, 2021, 190, 116681 Water Research). Please consider this suggestion during the revision.

> We agree that a conceptual diagram would more clearly link the predictions of the model to potential ecological consequences. We will add the diagram as Figure 5 and refer to it in the Discussion. In addition, we add a section to the Discussion that outlines the consequences of the absolute changes as well. We would not include the papers suggested by the reviewer in this section, because Hebert et al. attribute most of their findings to a later start of ice-on (while we focus on ice-off) and Kong et al. point to warmer water temperatures promoting winter (i.e. pre-spring) blooms, but our model did not simulate winter blooms and observations indeed suggest that these are uncommon in Lake Erken (at least in comparison to the magnitude of the spring blooms), perhaps due to the larger depth or more intense winters compared to that study.
>
> New figure:

[Figure]

Caption Figure 5. Conceptual diagram of how the changes in spring event timing in Lake Erken under a warming climate, as simulated by the GOTM-WET and SWAT+ models, would link to ecological consequences in the lake.

*Add after L. 224:* These absolute changes in the timing of spring events may lead to several changes in lake state (Figure 5). For instance, the earlier spring chlorophyll peak leads to an earlier uptake of nutrients and a longer growing season. However, not all effects are restricted to spring itself. An earlier onset of stratification leads to lower hypolimnetic oxygen concentrations in the summer (Jane et al., 2023) and a longer period of nutrient limitation in the epilimnion (Sommer et al., 2012). These trends were indeed seen in our future projections, with chlorophyll and epilimnetic nitrate concentrations remaining constant despite increased nutrient loading (Jiménez-Navarro et al., 2023), and were likely partially driven by the earlier stratification onset.

- 2B It is confusing to learn that the model did not catch the actual dynamics in certain years (Fig. 2). Despite the reason of the methodology or definition of the events, it would be necessary to provide an acceptable explanation for these 'bad' years not only in the supplements but also in the main text. For example, are these bad years featured by hydrological or climatic extremes? or there were malfunctions of the sampling infrastructure?

We will expand the text on the potential causes of the bad years, with a particular intent to see if there were consistent patterns that were missed by the model. One important addition, therefore, was that the ice-off date was simulated significantly too late in years that had very little ice cover, and this may have implications for our future projections as well. However, we prefer to avoid discussing individual years in the main text, so we retain the discussion of individual years in the Supplement (Figure S5, Table S1). As the reviewer argues, it would indeed be interesting to know if there are consistent features that are causing the model to underperform, and we wanted the text to reflect this, rather than discussing peculiarities of certain years.

L. 155: Particularly, this occurred in years with short ice cover duration, in which the 2 °C threshold may estimate ice-off to occur too late.

*Add after L. 205:* The method to estimate ice-off from the model results (a 2 °C threshold) tended to simulate ice-off too late in years with low ice cover. Therefore, our study is likely underestimating the advancement rate of ice-off date, and ice may be disappearing even faster than the rates predicted here.

*Change to Table S1: indicate as annotation for years 2008, 2014, 2020, and 2021, for ice-off, that these years had the lowest recorded ice duration in the study period.*

As said, bad fits of ice-off tended to occur in years with little ice, in which the 2-°C threshold did not prove accurate (though surface water temperature was simulated well), as factors like wind may play a bigger role. We could not find consistent patterns in bad fits of discharge or spring chlorophyll: as can be seen in Figure 2, anomalous years (late discharge in 2010; early chl peak in 2008) were sometimes simulated very well by the models and sometimes not at all (late discharge in 2013; early chl peak in 2000). For stratification, and this will now be further clarified in the text, the issue simply seems to be the noise in the observed data interfering with the threshold-approach to determine onset. We tested multiple thresholds, but due to observations being noisier than simulations, there were some consistent mismatches regardless of the choice of threshold value (despite such methods being well-established in literature). However, these issues are unlikely to have an effect under future climate, and in fact, as Figure S5 shows, bottom-top density difference early in the year was actually very well simulated by the model.

New L. 152-153: Upon this further inspection, we concluded that for the five badly simulated years for discharge and chlorophyll, the model did indeed not capture the dynamics of the lake or catchment, though without indication that particular events led to a systematic over- or underestimation.

L. 157: As such, we concluded that it was noise in water temperature observations that caused the threshold method to occasionally fail, rather than an inability of the model to simulate the state of the lake.

- 2C The mixture terminology of 'processes' and 'events' should be reconciled. Are there any differences? If not, please avoid switching terms and be consistent throughout the text. It would facilitate reading if all the 'events' changed to 'processes', or vice versa.

This is a valid comment and in line with comment 1C of Reviewer 1. When referring to the four spring events (ice-off, onset stratification, etc.), we will now consistently use the term "events".

Minor comments

- **2D** Abstract, please specify which 'process' are referring here at the very beginning (e.g. eco-hydrological processes).

> We will now specify that this relates to physical and biogeochemical events
>
> New L. 13-14: Lakes experience shifts in the timing of physical and biogeochemical events as a result of climate warming, and especially relative changes in the timing of events may have important ecological consequences.

- **2E** Line 45, if these processes are well acknowledged to be interlinked and occurs in causality and order already, what is the rationale to study them together? I think it should be further stressed that these processes have different sensitivity to climate change, and may response asynchronously in the future with changing orders and causal linkage. As a result, we must evaluate them together.

> We discuss why we may expect to see asynchronous changes in the final paragraph of the Introduction, and we will further clarify and expand this section.
>
> New L. 52-64: We used a coupled catchment-lake model framework to make future projections of the timing of these four events (ice-off, spring discharge, the spring phytoplankton bloom, and onset of stratification) and additionally to compare the projected trends between each of them. The use of process-based models can provide a robust framework for future projections of the timing of these springtime events, and the numerical coupling of lakes to their catchment allows a more thorough evaluation of climate change impacts and environmental changes (Kong et al., 2022). We hypothesised that all events would occur earlier in the year in a future, warmer climate, which is in line with previous studies, but also that relative changes in the timing of these events would occur. The latter expectation was partially due to the different processes driving each event, for example early-spring rain and air temperature would have the greatest importance in affecting snow and ice melt, while wind and temperature later in the season would affect the onset of stratification. Moreover, the effect of the strong seasonal cycle of solar radiation at the latitude of our study site would provide different physical constraints on phytoplankton, stratification, ice-off, and discharge. Climate warming could therefore affect not only the timing of these events, but also how they depend on each other and other external forcing – for example, in a future climate, the spring phytoplankton bloom might no longer rely on ice-off, but on the seasonal increase in solar radiation. The aim of our study is to create future projections of the timing of ice-off, spring discharge, the spring phytoplankton bloom, and onset of stratification and assess their absolute and relative changes, in order to better understand the impact of climate change on springtime events in lakes.

- **2F** Line 52-53, what are the four processes? Please specify, or define them earlier with a clear name and use this name thoughout the text.

> We will add the four events between brackets.

L. 52: …these four events (ice-off, spring discharge, the spring phytoplankton bloom, and onset of stratification)

- 2G Please summarize the main hypotheses and/or research questions with bullets by the end of introduction section.

We prefer to refrain from the use of bullet points (in line with other studies in this journal), but we will rewrite the (last paragraph of the) Introduction, and close off this section by re-stating the aim, as we acknowledge that previously, the last part of the Introduction was rather open-ended. Our hypotheses will also be found in this paragraph.

*See our reply to comment 2E for the new last paragraph of the Introduction*

- 2H Please increase the font size in Figure 2.

The small font size was an oversight, for which we apologise. This will be improved in the new Figure 2, which will include some additional changes in response to Reviewer 1's comment 1M.

New Figure 2:

[Figure]

Caption Figure 2. Simulated (red diamonds) and observed (black circles) timing of (a) ice-off, (b) 50% cumulative spring runoff, (c) spring chlorophyll peak, and (d) onset of stratification. The years are on the y-axis, and the difference in timing is shown by a dashed line. The units on the x-axis are in day-of-year (DOY). The light grey area

indicates the validation period. Open diamonds denote the years that were fitted badly (> 14 days error) and that are further investigated in Supplement section S2.

- **2I** Figure 4, if I understand correctly, the color represents the 'ratio of the slope', rather than the slope itself. Please correct the title of the legend bar to avoid any confusion.

The colours do represent the Sen's slope itself, but of one event relative to another. So they are Sen's slopes fitted on "$DOY_{spring\_chl} - DOY_{ice-off}$", "$DOY_{spring\_chl} - DOY_{onset\_strat}$", etc. This was done instead of showing a ratio, because this way, p-values could be computed. We will add an extra example to the caption of Figure 4.

Caption Figure 4. … on the y-axis. For example, under SSP 5-85, ice-off date advanced faster than the date of the 50% discharge…

- **2J** Line 212, it is intriguing to see that the onset of stratification is always later than the Chl-a peak event, even at the very beginning of the simulation in 1985 (Fig. 3). Conventionally, as already stated in the introduction, the onset of stratification is a prerequisite for the spring phytoplankton bloom (Line 49). Are there any observations in Lake Erken, that the current or previous spring phytoplankton blooms were already earlier than the onset of stratification since 1985? Are these species diatom, according to field data and model predictions? Overall, it is necessary to add a few more explanations here.

Although the maximum depth of 21 m might suggest a reliance of phytoplankton growth on stratification, the lake's mean depth is only 9 m and before the spring bloom, the water is rather clear (typically >4 m Secchi depth in winter and low coloured dissolved organic matter). As such, stratification is not a prerequisite for phytoplankton growth in Lake Erken, at least not for the diatoms that dominate the spring bloom. In L. 49, we mentioned that stratification is a prerequisite under turbulent conditions in deep lakes (though we acknowledge that water clarity plays a role as well, and that it could be hard to provide a threshold value between "shallow" and "deep" in this regard).

Both in response to this comment and comment 1N, we clarify the sentence and refer to Figure 2 and two references. We will add as well that this spring peak is dominated by diatoms, as shown by Weyhenmeyer et al. (1999).

L. 215: In Lake Erken, spring phytoplankton growth is not reliant on stratification due to the limited mean depth of the lake, and the spring chlorophyll peak (dominated by diatoms) tends to occur prior to onset of stratification (Figure 2; Weyhenmeyer et al., 1999; Moras et al., 2019), …

Additional references, not previously cited in manuscript:

Jane, S. F., Mincer, J. L., Lau, M. P., Lewis, A. S. L., Stetler, J. T., and Rose, K. C.: Longer duration of

seasonal stratification contributes to widespread increases in lake hypoxia and anoxia,

Global Change Biology, 29, 1009-1023, 10.1111/gcb.16525, 2023.

---

## Author Comment (AC3)

In this manuscript the authors use a one way coupled drainage area to lake model setup to investigate future climate impact on spring processes including ice-off, 50% cumulative spring discharge, spring phytoplankton bloom and stratification onset. The bold and novel model setup include stream flow, nutrients and temperature (SWAT+, LOADEST, air2water) coupled to lake physics and biogeochemistry (GOTM, WET). The important findings of the authors show how the occurrence of important spring processes are occurring earlier in a future warmer climate. The manuscript is in a good order but would benefit from extra clarity, sliming down and expansion as my points hereunder show.

> We thank the reviewer for the thorough assessment and the requested clarifications, which allowed us to improve the manuscript. We respond to each comment below, with our responses in text boxes and proposed additions to the text in red.

3A This manuscript continue and analyze deeper the effect of climate from the work done in Jiménez-Navarro et al. (2023). The reader needs to clearly understand what is the difference between the two works, both in regard to which questions are being addressed here as well as be given all relevant information for spring processes. This point runs throughout the rest of this review.

> We thank the reviewer for this comment, and realised that there was indeed no statement at the start of the model framework description that stated this study as a continuation of Jiménez-Navarro et al. This will now be added. The previous paper described the setup of the models, the overall model performance, and the overall results of future climate projections. The present paper used the same model setup and simulations to look at spring events and how their timing might change under future climate conditions. We will add a line to clarify this.
>
> L. 77: The present study builds upon a coupled catchment-lake model setup created by Jiménez-Navarro et al. (2023). This model setup was used to simulate catchment discharges, nutrient loads, and in-lake conditions under present and future conditions, and in the present study, we additionally assessed simulations of spring events.
>
> We are aware that it can be challenging for readers to have information on the model in two separate places, and we strived to present (and if necessary, repeat) all necessary information for spring events in the current paper. The setup in two separate manuscripts was based on practical reasons, as we felt that a single paper with model descriptions, model performance statistics, future predictions, and additional analysis of spring events, would become too long and cluttered. We hope that our changes, to this comment and those below, will clarify the differences between the two studies.

3B The method description need to be expanded and put in line with Jiménez-Navarro et al. (2023). Among other things I cannot see how many parameters was used in air2stream, which is not a statistical model, it is a semi deterministic model (a hybrid process-based and data-driven model). Additionally, more detailed information regarding the GOTAM-WET model coupling is required. One of the things I miss is how transparency in the lake is modeled/treated. Do the biological model adjust lake transparency, and how do this affect spring bloom and stratification onset? And how do

the coupled model preform at deeper depth? The reader can now only see what happens at 3 m depth.

We used air2stream with 8 parameters. This information will be added.

L. 81: … and air2stream (8-parameter version, Toffolon and Piccolroaz, 2015; Piccolroaz et al., 2018),…

Yes, model components from the WET model (inorganic matter, particulate organic matter, and phytoplankton biomass) contribute to the turbidity. We will add this information. We did not run tests with and without biological feedback regarding transparency. However, the final value for the transparency parameter "g2" was very high (5.62 m, giving an extinction coefficient of only 0.18 $m^{-1}$, this information can be found in the Supplement of Jiménez-Navarro et al., 2023), therefore a very clear water column without considering the WET components. In the final model, the biological components therefore contributed a lot to the turbidity of the water column (Secchi depth varied between 6 m and 1 m), which is in line with observations in Erken (clear water – Secchi depth > 4 m - outside of the growing season, more turbid during the spring and summer blooms, when Secchi depths of 1.5 – 3 m are common).

L. 85: Light absorption by components in the WET model (inorganic matter, particulate organic matter, and phytoplankton biomass) feeds back to the physical model.

Performance at deeper depth is also reported in the Supplementary Material of Jiménez-Navarro et al., (2023). We will add plots of the simulated model variables at 15 m depth to the Supplement (section S1, Figure S1). However, we decided to not give further information about performance at deeper depth in the text. Despite its importance for lake dynamics, our focus is on spring events, when the lake is ice-covered, fully mixed, or starting to stratify. In this period, profiles are mostly homogeneous, with the only clear exception being the onset of stratification, and this is the latest event studied here. As such, we felt that reporting on the performance at deeper depth in the main text would not be in line with the focus of the present study.

New Figure S1:

[Figure]

Caption Figure S1. Time series of modelled (black line) and observed (blue dots for calibration, red dots for validation period) GOTM-WET variables.

3C the authors struggle with model correctness, needing to use a surface temperature threshold for ice-off despite having an ice module and need to explain discrepancies in stratification onset and chlorophyll spring peak. I ask myself how this can be and have some points here which might enlighten the manuscript. First do the grid resolution compared to measurement resolution affect the results? From Figure S5 timing of stratification it looks like the vertical lines denoting stratification onset do not match the

data and should in fact be earlier for the measurements (red line crossing green threshold before timing of stratification). Is this due to a too short window for continuing stratification, is there an error in the script, or do the resolution play a role? As for data. The one way coupled catchment and lake model setup was calibrated from 2000 to 2015 for the lake part and from 2007 to 2015 for the river part. Is the difference in calibration period affecting the results? Looking at Figure 2 for Ice-off this looks to be the case. And how do you deal with the 2000 to 2006 period in regard to river input into the lake model? Building on this, can the less than ideal  model correctness be explained by the location of measurements in and above the lake? Lake measurements come from a station at the deepest point in the lake ca 400 m from the eastern shore. This distance might be far enough away for near-shore processes to play a role, but are the location representative for the overall lake physics covering the central parts of the lake? Additionally but not required for this manuscript, it would have improved the results if the complete time frame of available data was considered for calibration, validation (if deemed necessary) could have been carried out in the start and not the end of available measurements see ex. Shen, H., Tolson, B. A. & Mai, J. Time to Update the Split-Sample Approach in Hydrological Model Calibration. Water Resour. Res. 58, (2022). https://doi.org/10.1029/2021WR031523.

We will treat the points raised by the reviewer separately.

We attribute the issues with the ice module to the lack of snow parameterisation in GOTM (L. 108-111). Although onset of ice is not affected by this, the offset (as predicted by the ice module) is likely to occur too early due to the lack of insulation that the snow provides. This is unfortunate, and an ice module including snow (such as Simstrat's) would have been better, but the choice for GOTM enabled the use of the WET model and its elaborate description of biogeochemical processes. We used the temperature-threshold to compensate for this issue, although it is not optimal (as explained further below as well).

The discrepancies in stratification onset had indeed to do with the time windows and density thresholds. Although the model simulates bottom-top density difference accurately most of the years (as seen in Supplement figure S5), the observed data is noisier than the modelled data, and the threshold approach occasionally defined different periods as the onset. We tested multiple thresholds (both regarding the time window and density difference), but there were some consistent mismatches regardless of the choice of threshold value (despite such methods being well-established in literature). Still, our method is rather well-established in literature and these issues are unlikely to have an effect for the climate projections, as the degree of noise in the signal will be constant in the model. Also in reply to Reviewer 2's comment 2B, we will clarify this further in the text.

L. 157: As such, we concluded that it was noise in water temperature observations that caused the threshold method to occasionally fail, rather than an inability of the model to simulate the state of the lake.

The shorter calibration period for the discharge was due to lack of measured discharge data before 2007. We do not understand the comment about Figure 2 in connection to the shorter calibration period for discharge, as the discharge did not seem to have a worse fit during the validation period. Regarding ice-off, we do indeed see a worse performance in 2020 and 2021. Rather than attributing this to the validation period, we expect this had to do with the exceptionally (though in coming

decades perhaps normal) short period of ice cover in those years. The temperature-threshold approach seemed to perform less well in such years. We will add this information to the manuscript. Moreover, since years with short or no ice cover will become more frequent, we will also add a line to the Discussion how this may impact our future projections.

L. 155: Particularly, this occurred in years with short ice cover duration, in which the 2 °C threshold may estimate ice-off to occur too late.

*Add after L. 205:* The method to estimate ice-off from the model results (a 2 °C threshold) tended to simulate ice-off too late in years with low ice cover. Therefore, our study is likely underestimating the advancement rate of ice-off date, and ice may be disappearing even faster than the rates predicted here.

*Change to Table S1: indicate as annotation for years 2008, 2014, 2020, and 2021, for ice-off, that these years had the lowest recorded ice duration in the study period.*

Although the shoreline is not too far away from the monitoring location, there are no major inflows anywhere near, as the largest part of the watershed is to the west of the lake. The measurement location is at the edge of the main basin of the lake, and for example seiche movements are occasionally visible in the high-frequency data (though they have a frequency around one day, so disappear with daily averaging). Regarding the processes under study here, we do not foresee a major effect of the location where the measurements were chosen, though we acknowledge that measurements from a single location are only moderately representative of the whole lake. Ice cover is likely longer in secluded bays compared to the main basin, stratification can form in shallow areas first (thermal bars), and blooms may occur in bays while the main basin is less affected, but overall, we expect our measurements to be moderately representative, due to the open connection to the main basin.

We appreciate the reviewer's comment regarding using the full period for calibration, with potential validation at the start. Although the reviewer did not request particular changes to be made to the manuscript, we will take the opportunity to elaborate on this issue, perhaps for no other reason than that we find it interesting as well! We wanted to use the same model setup as Jimenez-Navarro et al. for reasons of clarity, but in general, this is an interesting proposition and it could be considered whether the model would have been more accurate if we had considered all data for calibration. In our view, the degree to which models have been established and are prone to overfitting, plays a large role here. Hydrological and hydrodynamical models are based on purely physical equations that have been widely applied (even if the models themselves have not) and are usually not heavily calibrated, so that overfitting, or other issues related to model stability, are less of a big risk. In biogeochemical models, however - at least the rather complex type that we used here -, many parameters are calibrated, many different equations are in use that describe the same process, overfitting is a real risk, and pools of N or P running dry can easily lead to unrealistic projections. A separate validation period may help to partially countermeasure these issues. Likewise, a validation period at the start could have downsides if there is still an effect of initial conditions, as biogeochemical models may need a longer spin-up than physical models. So, in this sense, we wonder if the recommendations of Shen et al. could/should be extended to biogeochemical models. Yet at the same time, data availability (both in frequency and period of coverage) is more pressing for biogeochemical variables, so being able to use the full period for calibration would have additional benefits as well. In short, we consider this topic

outside the scope of our present study, but absolutely see the importance of looking further into this. We believe that the aquatic modelling community would benefit from an open discussion on this topic and indeed numerical testing of various methods, to find the advantages and limitations of different calibration and validation strategies.

3D Lake processes are heavily dependent on local atmospheric conditions, so to for the drainage area processes. The authors used five GCM models which by their global nature are course resolved. The GCMs are bias correction toward local measurements in Jiménez-Navarro et al. (2023), but if I understand Supplementary table E $4^{th}$ column (RMSE) this bias correction is almost nonexistent.  Taking the difference between GCM INM-CM5-0 and measurements as an example, mean air temperature RMSE (Root Mean Square Error) drops from the unbiased comparison of 5.712 °K to 5.687 °K after bias correction and for Wind Speed from 4.283 to 2.588 m/s, and improvement with <1% and ~40 % respectively. The bias correction of precipitation, a key input to the drainage area model, looks to have failed. Now I might misunderstand how the Bias correction results are shown, but this illustrate my first point. Can we trust that the calibration is still valid using the climate models as input? Additionally, the reader needs to know why these climate models and scenarios were selected. I suspect because they cover the extreme ranges of for example temperature, precipitation, wind speed etc.. Furthermore since the setup is used for projecting climate effects, is the time frame (for drainage area and lake) long enough so that the models capture the climate trend (which is small compared to seasonal variations)? It would help the reader to see how the trends during the setup/calibration period are in the model compared to measurements.

We would argue that our bias-correction succeeded: the quantile mapping method that was used only aims to decrease the bias. The Supplementary table in Jiménez-Navarro et al. (2023), referred to by the reviewer, indeed shows that Bias significantly decreased and, in many cases, also RMSE, though there were cases in which RMSE slightly increased. With regard to precipitation, the bias correction did not fail, but we admit that we should have used scientific notation to show the RMSE and Bias values, which now appear to be 0 due to the unit (kg/m2/s, or mm/s). For example, for GCM INM-CM5-0, the precipitation bias before correction was $-2.7 \cdot 10^{-6}$ mm/s, and after bias correction it was $-8.5 \cdot 10^{-7}$ mm/s. In some combinations of GCM and variable, the bias correction had indeed little effect, but only because the GCM prediction already was relatively unbiased compared to observations.

A high NSE and low RMSE cannot be expected, as a hindcasted GCM is not intended to simulate the same weather events as were observed (e.g. a storm may pass at a different time than observed), but it should rather reflect the observed weather over a longer temporal scale (as opposed to a reanalysis dataset, which does intend to match observations as close as possible). A biased GCM, however, would be an indication of a biased prediction, and it is this that the quantile mapping mitigated. We therefore don't consider this study to be less accurate in terms of its future projections than other studies, though of course these projections present a large degree of uncertainty.

The five GCMs were selected because they represented a wider range of predictions compared to a single projection, but another main reason was that these GCMs provided all the necessary forcing needed to run both SWAT+ and GOTM-WET. Some other GCMs, for example EC-Earth-Veg and GFDL-ESM4 that were included in an earlier study using SWAT+ in Lake Erken (Jiménez-Navarro et al. 2021, doi:

10.3390/f12121803), missed some variables that were needed to run GOTM-WET (at least without using a different approach compared to the other GCMs).

We will now state in the manuscript that these GCMs provided the required forcing and that they were bias-corrected.

L. 92: Each GCM provided the meteorological forcing required to run both SWAT+ and GOTM-WET and the projections were bias-corrected to locally observed meteorological data using quantile mapping (see Jiménez-Navarro et al., 2023).

As in many climate projection studies, the time frame with measurements is indeed comparatively small to detect climatic trends, and the projected simulations are longer than the period with measurements itself. To test whether our model detected climatic trends during the calibration and validation period, we selected several model output variables that were predicted to show a trend with warming in Jiménez-Navarro et al. (2023): discharge, water temperature, oxygen, and $NH_4$ concentration. Both 3- and 15-m depths were assessed and annual averages were taken, and for simplicity, gaps in observations were linearly interpolated in order to fit a Mann-Kendall test. At a 0.05 significance level, according to the Mann-Kendall test, only corresponding trends in 3- and 15-m simulated oxygen concentration could be found, and in the observations, none of the variables showed a similar trend as in the climate simulations. In short, the reviewer's question could therefore be answered with "No, the calibration/validation time frame is not long enough to capture a climate trend". It should be noted that over longer periods, climatic trends in historical data of Erken do become visible for physical parameters at least (see Moras et al., 2019, cited in manuscript). Still, we do not consider this a restriction for this study. Considering biogeochemical data, Lake Erken has a comprehensive dataset, covering a longer period than most other sites (even longer data is available, but less regular and less variables, which is why we do not model even further back). As such, for this type of studies, it presents an optimal site to do this, and a lack of climatic trends over a comparatively short period in both observations and model, does not invalidate the use of the model itself. Since these findings are more in line with the "general" model performance, we did not see a convenient place in the manuscript to add this information, as this would rather be added to the manuscript by Jiménez-Navarro et al. (2023). In the current paper, readers can see both observed and simulated spring event timing over the period with observed data in Figure 2. Nevertheless, we hope that this information satisfies the reviewer.

3E Through the analysis of trends from the climate simulations, the authors treat the climate scenarios as constant change over time ex. Fig 3. This is not correct, in fact the gradient for each scenario change over time, especially for SSP 245. I suggest dividing the model output into 30 year chunks while conducting the analysis, or look at the amount of change from a reference to a far future period.

Although the air temperature change is indeed not linear (at least for SSP 2-45), we chose a linear model because it fitted the response well (Figure 3). We retained the use of a linear model for ease of communication and to facilitate a comparison with previous studies, which report phenological trends often as well as "x days per decade". From a practical point of view, the Mann-Kendall analysis allowed us to also assess relative changes. However, we agree that reporting the output as suggested by the reviewer has benefits, and it could further facilitate comparison with other

studies and future meta-analyses. We now additionally report the values for the chunks 1985-2014, 2040-2069, and 2070-2099 in the Supplement and refer to these values in the Results and Discussion. It can indeed be seen there that some future trends do not seem to behave linearly, such as the chlorophyll peak date under SSP 2-45 (although it should be noted that the mid-century period is not in the middle of the other two periods). Additionally, we will shortly discuss linearity and how the slopes should be interpreted, in the Discussion.

*Add after L. 224, and after the changes made in response to comments 1O and 2A of the other reviewers:* Although the predicted changes in event timing are reported as linear trends, it should be noted that we do not assume that these changes are entirely linear. Especially in SSP 2-45, the development of air temperature through the simulation period is not linear, and the timing of events will not follow a linear trend over time either. In Supplement section S3, averages in separate time periods are reported, and for instance the advance in timing of the spring chlorophyll peak gives an indication of slowing down or stopping in the second half of the century under SSP 2-45. Reported linear changes should therefore be seen as the average change over the period 1985-2100, and we did not investigate the shape of the trend during this period.

New section in the Supplement:
**S3. Future projections - Time periods 1985-2014, 2040-2069, and 2070-2099**
Table S2. Average values for time periods 1985-2014, 2040-2069, and 2070-2099 under the SSP 2-45 and 5-85 scenarios.

| Variable | Unit | 1985-2014 | SSP 2-45 | | SSP 5-85 | |
| --- | --- | --- | --- | --- | --- | --- |
| | | | 2040-2069 | 2070-2099 | 2040-2069 | 2070-2099 |
| Chlorophyll peak date | DOY | 108.31 | 87.46 | 89.66 | 86.01 | 77.64 |
| Peak spring chlorophyll concentration | mg/m$^3$ | 14.53 | 11.71 | 11.22 | 11.81 | 10.84 |
| 50% spring discharge date | DOY | 78.37 | 60.43 | 59.13 | 57.52 | 55.31 |
| Cumulative spring discharge | m$^3$ | $8.92 \cdot 10^6$ | $1.10 \cdot 10^7$ | $1.16 \cdot 10^7$ | $1.20 \cdot 10^7$ | $1.29 \cdot 10^7$ |
| Ice-off date | DOY | 101.93 | 90.50 | 83.83 | 80.96 | 68.15 |
| Ice-on date | DOY | 3.79 | 28.01 | 35.03 | 34.21 | 45.24 |
| Number of days with ice | days | 72.02 | 31.31 | 24.16 | 21.06 | 7.04 |
| Average ice thickness | m | 0.155 | 0.061 | 0.048 | 0.039 | 0.014 |
| Stratification onset | DOY | 140.71 | 132.26 | 130.78 | 128.61 | 125.65 |
| End of stratification | DOY | 261.49 | 267.89 | 267.81 | 271.13 | 273.29 |
| Number of stratified days | days | 122.11 | 136.56 | 138.56 | 143.41 | 149.11 |
| Average Schmidt stability during stratification | J/m$^2$ | 177.71 | 221.87 | 232.13 | 236.61 | 266.61 |
| Average mixed layer depth during stratification | m | 6.53 | 6.35 | 6.32 | 6.00 | 6.01 |

---

## Author Response (AR1)

Dear authors

thanks for the detailed response to the three referees. Please implement the changes in the revised manuscript.

I have another comment from my own reading. You wrote on p5 "A comparison between simulated and observed inflow and lake data, spanning 2000-2021 for most variables, confirmed that the models reproduced the dynamics of the system with reasonable accuracy (see Jiménez-Navarro et al., 2023)". The term "reasonable accuracy" lacks a clear benchmark and is quite subjective. Can you explore the specific criteria that insure confidence in the model's performance? At some point, the community should converge toward a standard way to adress the performance of a model. Looking for instance at the TP or DO (on your new SI), it seems that the model has some difficulties to reproduce the observed dynamics. I recognize that the calibration/validation process primarily occurred in a prior study. However, I would like to see a deeper exploration of the specific metrics that would validate to the communty your confidence in the model's adequacy for addressing your research question (e.g. Timing of spring events changes under modelled future climate scenarios)

Dear Editor,

We thank you for the opportunity to submit a revised manuscript and we give a point-by-point reply to the reviewers' comments below, with our reply in boxes. All line numbers refer to the manuscript without tracked changes. The comments were very helpful, amongst others to clarify our findings for the reader.

In response to your comment about communicating our confidence in the model regarding spring event timing, we refer to the modified Section 3.1: though for chlorophyll and discharge, there are indeed a few years in which the model failed to reproduce the observed patterns, wrong predictions for ice-off and stratification onset relate more to the methods by which they are calculated. Despite testing different methods and thresholds, some years showed a bad fit of the event despite visual inspection showing overall good model performance (Supplement Section S3, see plots for ice-off and stratification onset). Our main reasoning regarding this latter issue is that this would not affect the future predictions made in our paper, as then trends in the model are assessed, rather than a comparison to observations. Thresholds and minimum durations in the calculations of ice-off and stratification onset were sometimes at odds with the complexity and noise inherent in environmental observations, but this would not be an issue in future model simulations.

Regarding your comments about a deeper exploration of the model performance, a standard way of addressing model performance, and the lack of clear benchmarks: we agree that the assessment of biogeochemical models is often rather subjective, and that the lack of a standard methodology is a barrier to improved model application. We therefore decided to amend our manuscript with a validation along the lines suggested by Hipsey et al. (2020, doi:10.1016/j.envsoft.2020.104697), which is one of the few papers that we are aware of that propose a standardised framework that can be applied to a wide range of biogeochemical models. We added a validation of several additional variables, exceeding the commonly-used "level 1a" validation (i.e. direct comparison between model results and observations) – which was performed in the Jiménez-Navarro et al. (2023) paper. Please see the new section S2 in the Supplement, and the added script in the workflow. These additional variables were chosen based on what output was provided by the model and what data were available. The Hipsey et al. framework does not provide thresholds for "good" or "acceptable" performance, which is difficult or even undesirable in case of biogeochemical models. For example, metrics are hard to determine or misleading for variables with spiky behaviour, such as phytoplankton blooms (see

Elliott et al., 2000, doi: 10.1016/S0304-3800(99)00221-5; Jachner et al., 2007, doi:10.18637/jss.v022.i08); an improved simulation of one variable can be at the expense of another; some of the most limiting (i.e. relevant) nutrients may operate close to detection limits; and for some dynamics (especially the "level 3" validation in the Hipsey paper), quantitative metrics cannot be determined (e.g. formation of a gyre or a deep chlorophyll maximum). A judgement of whether the model fits sufficiently well remains, therefore, subjective. However, as stated by Hipsey et al., by following this framework and stating the level at which the assessment was performed, modellers can communicate more clearly how validation was performed, how this influences model uncertainty, and promote comparability between studies.

Sincerely,
Jorrit Mesman, on behalf of all authors

**RC1**

General Comments

- Overall, this is a well-written and important contribution to our understanding of changing ecosystem functioning in lakes. It provides new insight by both developing projections of biogeochemical variables like phytoplankton dynamics and a novel comparison of the relative change in timing of important limnological events. The methods and results are clearly presented and the research is well contextualized. I suggest a few improvements below to better present the research in the context of other studies and research within the field of limnology.

> We thank the reviewer for the kind comments and for their useful suggestions, which allowed us to clarify and improve the manuscript. We respond to each comment below, with our responses in text boxes.

Specific Comments

- 1A If you're not limited by words already, consider adding a sentence in the abstract that states how well your model did during training/validation to add support to the validity of your projections

> We added a sentence related to the model performance (L. 19)

- 1B In the first few sentences of the intro, can you add some language to make it crystal clear whether the studies you are citing demonstrated *already observed* changes or projections in timing of processes? On a glance, I think most of the studies you cite are observed already and adding a short paragraph that more thoroughly summarizes findings from other projection studies would help highlight the novelty of your approach (including phytos and catchment loading and comparing relative changes in annual timing events across multiple variables)

> We added a sentence with additional projection studies to link past observations to future projections (L. 35).

- 1C I think you are using spring 'metrics', 'events', 'processes' interchangeably to refer to your four response variables in the intro—might be good to choose one and stick with that

> This is a valid comment and in line with comment 2C of Reviewer 2. When referring to the four spring events (ice-off, onset stratification, etc.), we now consistently use the term "events".

- **1D** You introduce some really good, but new, content in the last paragraph of the intro (line 59 on) about why relative differences in the timing of spring events matters. I wonder if you could make this its own paragraph before you introduce your study and hypotheses so that you can expand a bit more on why relative shifts in timing matter—this is the key finding from your study so it should be emphasized heavily in the intro

> We agree that it's beneficial to highlight the importance of relative shifts in timing more. We have added a new paragraph to the introduction and included some additional ways in which relative shifts could matter (L. 75-84). Moreover, we have restructured the last paragraph in response to comments from Reviewer 2, so that the Introduction ends with aims and hypotheses, instead of new content.

- **1E** You might want to add a citation in the introduction somewhere to Adrian et al. 2012 who discuss how changes in climate drivers during key time periods are critical to informing overall ecosystem function: Adrian, R., Gerten, D., Huber, V. *et al.*Windows of change: temporal scale of analysis is decisive to detect ecosystem responses to climate change. *Mar Biol* **159**, 2533–2542 (2012). https://doi.org/10.1007/s00227-012-1938-1

> This is indeed a relevant reference for the importance of critical events and their effects later in the season; thank you for this suggestion! It is now cited in L. 39, L. 42, and L. 48.

- **1F** Line 42: I suggest remove 'in this study' phrasing and focus on why these metrics are important generally in this paragraph before you emphasize the details of your study specifically

> We removed the phrase "In this study" (L. 58) and added several sentences to the Introduction that outline the importance of the four events under study (L. 60-66).

- **1G** Methods, line 97-102: can you provide reference to any other studies which use biogeochemical process models and have similar $R^2$ for reproducing observations? I'm not implying that the fit isn't good enough, just that comparing to what others have done would be helpful to justify some of the lower $R^2$ values

> We have now added references to three studies (Chen et al., 2020; Kong et al., 2022; Zhan et al., 2023, cited in manuscript), which report similar goodness-of-fit for biogeochemical variables, also using coupled physical-biogeochemical models (L. 147-149).

- **1H** Line 108: can you provide a date range for the historical record of ice-off dates? Also in this section, can you report the bias for your 2C threshold for simulating ice-off for comparison since you report the error using the ice module?

A date range is now provided for ice-off dates (L. 157-158) and the MAE and ME are now also reported for the ice-off calculation with the temperature threshold (L. 163).

- 1I Line 145: include a citation for your workflow here as well?

Thank you for this suggestion: we added the citation for the workflow (L. 198-199).

- 1J Line 148-149: a sentence similar to this would add strength to the abstract in demonstrating that your model performed well against observations. I would suggest adding the years of this calibration/validation time period here (not necessary for abstract though I think)

In line with comment 1A, we added a sentence to the abstract (L. 19).

We decided to not split the results between calibration and validation period in the text. The percentage of events with an error less than 10 days was in fact the same for calibration and validation, but as can be seen in Figure 2, there were clear differences between the variables, so such a statement might give a wrong impression. Moreover, the number of years was necessarily limited and the validation period contained a small number of years, so we preferred to report the results for the whole period together. Figure 2 allows the reader to assess each variable and period separately.

- 1K Results, line 170: maybe just me, but I'm not familiar with the term 'shoal'. Could you rephrase as 'increase' or 'decrease'?

The term is indeed not used often; we now use "become shallower" instead (L. 231).

- 1L Table 1: Is there a way you could visualize this rather than providing a table (but perhaps keep table in SI)? I'm envisioning something similar to Figure 2 where you show the difference between the value at the beginning of the simulation (intercept) and the mean value at the end of the projection time period based on Sen's slope? This would allow you to highlight the directionality and magnitude of average change

We agree that a visualisation is often more intuitive than a table for readers, but we preferred to leave the table as is, because a) the table contains a large number of variables, and not all are the main focus of the paper, and b) each variable has different units, which would complicate a figure.

- 1M Figure 2: can you make the font size overall a bit larger? It is necessary for me to zoom in quite a bit to read it as is. Would also suggest adding panel labels if this is a journal requirement. Instead of the purple square, maybe could you make the red

diamonds open for years with a bad fit, filled for years with a good fit? The square is a little distracting (not a major issue though)

> We were indeed struggling with how to represent the badly-fitted years in the plot, and we really liked the idea of using open and closed diamonds for this. Thank you for this suggestion! We additionally added panel labels.
>
> The small font size was an oversight, for which we apologise. This has been improved as well in the new Figure 2. The script in the workflow has been updated accordingly.

- **1N** Line 215: is there a figure you can reference to support this? As it's written, it's unclear if you mean under current conditions or under future projections

> Also in response to comment 2J, we have clarified the sentence and refer to Figure 2 and two references (Weyhenmeyer et al. 1999 and Moras et al., 2019, cited in manuscript) (L. 284-286).

- **1O** Line 245: this is really interesting. Did you calculate chlorophyll-a concentrations later in the growing season or just spring? I am wondering if there is an antecedent effect for later in the year which could have broader implications for additional bloom events and could be useful to add to the discussion

> We focus in our study on the spring period, but have model output for the whole year. If the lower spring peak was indeed caused by a gradual shift to more light limitation compared to nutrient limitation (L. 291-292), while nutrient inputs stayed the same, one could expect a "broader" rather than "higher" spring peak, and this could have implications later in the season. It would be difficult to assess whether there would be causal links in our current model setup, however, and this would require additional experiments that would distract from the main message in the paper. However, also in response to Reviewer 2's comment 2A, we have added a section to the Discussion on effects of spring events beyond spring itself (L. 297-303).

- **1P** Line 272: I think you should emphasize that this is especially true for biological responses like chla (there are studies looking at multiple connected hydrodynamic processes, Ayala et al. 2020, Barbosa et al. 2021, Feldbauer et al. 2022, Desgue-Itier et al. 2023, Wynne et al. 2023, etc.)

> We agree, and have added a phrase that stresses the novelty of also looking into biological and watershed responses (L. 364-366). The most notable exception of biological processes studied in conjunction would be phytoplankton-zooplankton phenology, which may not have been apparent in our Introduction, so we added a sentence about this as well (L. 55-56).

- 1Q I think the study could benefit from more discussion of the implicit assumptions from focusing on spring event timing as your response variables (e.g., instead of summer, winter, or fall events). You do a good job justifying why spring is important (and I believe it), but I think you could add context which highlights other research which shows that antecedent conditions during other time periods (e.g., winter-time dynamics, storm events) are important for year-round functioning and adding some context to acknowledge this would be helpful in the discussion. Some potential citations

  o Cavaliere et al. 2021 https://doi.org/10.1029/2020JG006165

  o Adrian et al. 2012 https://doi.org/10.1007/s00227-012-1938-1

  o Thayne et al. 2021: https://aslopubs.onlinelibrary.wiley.com/doi/full/10.1002/lno.11859

We have added context regarding critical timing windows and antecedent conditions in other parts of the year, as suggested. We mention storms, the occurrence of autumn blooms, and the effects of incomplete mixing in winter. We mention this in the Introduction (L. 41-47), before going deeper into springtime events.

RC2

The present study elaborated to investigate the future climate impacts on the spring hydrological and ecological processes (i.e. spring discharge, ice-off, spring phytoplankton peak, onset of stratification) in a typical temperate lake Erken. The findings have critical implications because these processes were rarely evaluated simultaneously and their different sensitivities to climate change may result in different change paces or rates, and eventually lead to profound consequences on lake ecosystem in the future. The paper is well-prepared and concisely written. I have a few major and minor concerns and would like to recommend publication if the authors can address them properly during the revision.

> We appreciate the compliments and also the concerns and suggestions raised by the reviewer. We respond to each comment below, with our responses in text boxes.

Major comments:

- **2A** The manuscript repeatedly emphasizes the ecological consequences of the different rate of advancing among the four investigated events, but these are not actually evaluated and subject to inferences and speculation, which may be attributed to the limitation in the model. This can be compensated, and manuscript can be improved, if the author can add a conceptual diagram in the discussion section, which summarizes the findings from the study (i.e. different advancing rate among the processes, increasing gap between stratification onset and other processes, and the potential ecological consequences from literature, for example, increasing magnitude of winter diatom blooms, see e.g., Hebert et al., 2021 e2114840118 PNAS , or Kong et al, 2021, 190, 116681 Water Research). Please consider this suggestion during the revision.

> We agree that a conceptual diagram would more clearly link the predictions of the model to potential ecological consequences. We have added Figure 5 and refer to it in the Discussion. In addition, we have added a section to the Discussion that outlines the consequences of the absolute changes as well (L. 297-303). We did not include the papers suggested by the reviewer in this section, because Hebert et al. attribute most of their findings to a later start of ice-on (while we focus on ice-off) and Kong et al. point to warmer water temperatures promoting winter (i.e. pre-spring) blooms, but our model did not simulate winter blooms and observations indeed suggest that these are not common in Lake Erken (at least in comparison to the magnitude of the spring blooms), perhaps due to the larger depth or more intense winters compared to that study.

- **2B** It is confusing to learn that the model did not catch the actual dynamics in certain years (Fig. 2). Despite the reason of the methodology or definition of the events, it would be necessary to provide an acceptable explanation for these 'bad' years not only in the supplements but also in the main text. For example, are these bad years featured by hydrological or climatic extremes? or there were malfunctions of the sampling infrastructure?

We expanded the text on the potential causes of the bad years (L. 208-209, L. 211-212, L. 215-216), with a particular intent to see if there were consistent patterns that were missed by the model. One important addition, therefore, was that the ice-off date was simulated significantly too late in years that had very little ice cover (L. 211-212, Table S2), and this may have implications for our future projections as well (L. 270-273). However, we preferred to avoid discussing individual years in the main text, so we retained the discussion of individual years in the Supplement (Figure S5, Table S2). As the reviewer argues, it would indeed be interesting to know if there are consistent features that are causing the model to underperform, and we wanted the text to reflect this, rather than discussing peculiarities of certain years.

As said, bad fits of ice-off tended to occur in years with little ice, in which the 2 °C threshold did not prove accurate (though surface water temperature was simulated well), as factors like wind may play a bigger role. We could not find consistent patterns in bad fits of discharge or spring chlorophyll: as can be seen in Figure 2, anomalous years (late discharge in 2010; early chl peak in 2008) were sometimes simulated very well by the models and sometimes not at all (late discharge in 2013; early chl peak in 2000). For stratification, and this is now further clarified in the text (L. 215-216), the issue simply seems to be the noise in the observed data interfering with the threshold-approach to determine onset. We tested multiple thresholds, but due to observations being noisier than simulations, there were some consistent mismatches regardless of the choice of threshold value (despite such methods being well-established in literature). However, these issues are unlikely to have an effect under future climate, and in fact, as Figure S5 shows, bottom-top density difference early in the year was actually very well simulated by the model.

- 2C The mixture terminology of 'processes' and 'events' should be reconciled. Are there any differences? If not, please avoid switching terms and be consistent throughout the text. It would facilitate reading if all the 'events' changed to 'processes', or vice versa.

This is a valid comment and in line with comment 1C of Reviewer 1. When referring to the four spring events (ice-off, onset stratification, etc.), we now consistently use the term "events".

Minor comments

- 2D Abstract, please specify which 'process' are referring here at the very beginning (e.g. eco-hydrological processes).

We now specify that this relates to physical and biogeochemical events (L. 13).

- 2E Line 45, if these processes are well acknowledged to be interlinked and occurs in causality and order already, what is the rationale to study them together? I think it should be further stressed that these processes have different sensitivity to climate

change, and may response asynchronously in the future with changing orders and causal linkage. As a result, we must evaluate them together.

> We discuss why we may expect to see asynchronous changes in L. 90-97, and we have further clarified and expanded this section.

- **2F** Line 52-53, what are the four processes? Please specify, or define them earlier with a clear name and use this name thoughout the text.

> We added the four events between brackets (L. 85).

- **2G** Please summarize the main hypotheses and/or research questions with bullets by the end of introduction section.

> We preferred to refrain from the use of bullet points (in line with other studies in this journal), but we have rewritten the (last paragraph of the) Introduction, and now close off this section by re-stating the aim, as we acknowledge that previously, the last part of the Introduction was rather open-ended. Our hypotheses can also be found in this paragraph.

- **2H** Please increase the font size in Figure 2.

> The small font size was an oversight, for which we apologise. This has been improved in the new Figure 2.

- **2I** Figure 4, if I understand correctly, the color represents the 'ratio of the slope', rather than the slope itself. Please correct the title of the legend bar to avoid any confusion.

> The colours do represent the Sen's slope itself, but of one event relative to another. So they are Sen's slopes fitted on "$DOY_{spring\_chl} - DOY_{ice-off}$", "$DOY_{spring\_chl} - DOY_{onset\_strat}$", etc. This was done instead of showing a ratio, because this way, p-values could be computed. We have added an extra example to the caption of Figure 4.

- **2J** Line 212, it is intriguing to see that the onset of stratification is always later than the Chl-a peak event, even at the very beginning of the simulation in 1985 (Fig. 3). Conventionally, as already stated in the introduction, the onset of stratification is a prerequisite for the spring phytoplankton bloom (Line 49). Are there any observations in Lake Erken, that the current or previous spring phytoplankton blooms were already earlier than the onset of stratification since 1985? Are these species diatom, according to field data and model predictions? Overall, it is necessary to add a few more explanations here.

Although the maximum depth of 21 m might suggest a reliance of phytoplankton growth on stratification, the lake's mean depth is only 9 m and before the spring bloom, the water is rather clear (typically >4 m Secchi depth in winter and low coloured dissolved organic matter). As such, stratification is not a prerequisite for phytoplankton growth in Lake Erken, at least not for the diatoms that dominate the spring bloom. In the old L. 49 (new L. 71), we mention that stratification is a prerequisite under turbulent conditions in deep lakes (though we acknowledge that water clarity plays a role as well, and that it could be hard to provide a threshold value between "shallow" and "deep" in this regard).

Both in response to this comment and comment 1N, we have clarified the sentence and refer to Figure 2 and two references (Weyhenmeyer et al., 1999, and Moras et al., 2019, cited in manuscript) (L. 284-286). We added as well that this spring peak is dominated by diatoms, as shown by Weyhenmeyer et al. (1999).

**RC3**

In this manuscript the authors use a one way coupled drainage area to lake model setup to investigate future climate impact on spring processes including ice-off, 50% cumulative spring discharge, spring phytoplankton bloom and stratification onset. The bold and novel model setup include stream flow, nutrients and temperature (SWAT+, LOADEST, air2water) coupled to lake physics and biogeochemistry (GOTM, WET). The important findings of the authors show how the occurrence of important spring processes are occurring earlier in a future warmer climate. The manuscript is in a good order but would benefit from extra clarity, sliming down and expansion as my points hereunder show.

> We thank the reviewer for the thorough assessment and the requested clarifications, which allowed us to improve the manuscript. We respond to each comment below, with our responses in text boxes.

**3A** This manuscript continue and analyze deeper the effect of climate from the work done in Jiménez-Navarro et al. (2023). The reader needs to clearly understand what is the difference between the two works, both in regard to which questions are being addressed here as well as be given all relevant information for spring processes. This point runs throughout the rest of this review.

> We thank the reviewer for this comment, and realised that there was indeed no statement at the start of the model framework description that stated this study as a continuation of Jiménez-Navarro et al. This has now been added (L. 115). The previous paper described the setup of the models, the overall model performance, and the overall results of future climate projections. The present paper used the same model setup and simulations to look at spring events and how their timing might change under future climate conditions. We added L. 117 to clarify this.
>
> We are aware that it can be challenging for readers to have information on the model in two separate places, and we strived to present (and if necessary, repeat) all necessary information for spring events in the current paper. The setup in two separate manuscripts was based on practical reasons, as we felt that a single paper with model descriptions, model performance statistics, future predictions, and additional analysis of spring events, would become too long and cluttered. We hope that our changes, to this comment and those below, clarified the differences between the two studies.

**3B** The method description need to be expanded and put in line with Jiménez-Navarro et al. (2023). Among other things I cannot see how many parameters was used in air2stream, which is not a statistical model, it is a semi deterministic model (a hybrid process-based and data-driven model). Additionally, more detailed information regarding the GOTAM-WET model coupling is required. One of the things I miss is how transparency in the lake is modeled/treated. Do the biological model adjust lake transparency, and how do this affect spring bloom and stratification onset? And how do the coupled model preform at deeper depth? The reader can now only see what happens at 3 m depth.

We used air2stream with 8 parameters. This information has been added in L. 121.

Yes, model components from the WET model (inorganic matter, particulate organic matter, and phytoplankton biomass) contribute to the turbidity. We added this information to L. 125-127. We did not run tests with and without biological feedback regarding transparency. However, the final value for the transparency parameter "g2" was very high (5.62 m, giving an extinction coefficient of only 0.18 $m^{-1}$, this information can be found in the Supplement of Jiménez-Navarro et al., 2023), therefore a very clear water column without considering the WET components. In the final model, the biological components therefore contributed a lot to the turbidity of the water column (Secchi depth varied between 6 m and 1 m), which is in line with observations in Erken (clear water – Secchi depth > 4 m - outside of the growing season, more turbid during the spring and summer blooms, when Secchi depths of 1.5 – 3 m are common).

Performance at deeper depth is also reported in the Supplementary Material of Jiménez-Navarro et al., (2023). We added plots of the simulated model variables at 15 m depth to the Supplement (section S1, Figure S1). However, we decided to not give further information about performance at deeper depth in the main text. Despite its importance for lake dynamics, our focus is on spring events, when the lake is ice-covered, fully mixed, or starting to stratify. In this period, profiles are mostly homogeneous, with the only clear exception being the onset of stratification, and this is the latest event studied here. As such, we felt that reporting on the performance at deeper depth would not be in line with the focus of the present study.

3C the authors struggle with model correctness, needing to use a surface temperature threshold for ice-off despite having an ice module and need to explain discrepancies in stratification onset and chlorophyll spring peak. I ask myself how this can be and have some points here which might enlighten the manuscript. First do the grid resolution compared to measurement resolution affect the results? From Figure S5 timing of stratification it looks like the vertical lines denoting stratification onset do not match the data and should in fact be earlier for the measurements (red line crossing green threshold before timing of stratification). Is this due to a too short window for continuing stratification, is there an error in the script, or do the resolution play a role? As for data. The one way coupled catchment and lake model setup was calibrated from 2000 to 2015 for the lake part and from 2007 to 2015 for the river part. Is the difference in calibration period affecting the results? Looking at Figure 2 for Ice-off this looks to be the case. And how do you deal with the 2000 to 2006 period in regard to river input into the lake model? Building on this, can the less than ideal  model correctness be explained by the location of measurements in and above the lake? Lake measurements come from a station at the deepest point in the lake ca 400 m from the eastern shore. This distance might be far enough away for near-shore processes to play a role, but are the location representative for the overall lake physics covering the central parts of the lake? Additionally but not required for this manuscript, it would have improved the results if the complete time frame of available data was considered for calibration, validation (if deemed necessary) could have been carried out in the start and not the end of available measurements see ex. Shen, H., Tolson, B. A. & Mai, J. Time to Update the Split-Sample Approach in Hydrological Model Calibration. Water Resour. Res. 58, (2022). https://doi.org/10.1029/2021WR031523.

We will treat the points raised by the reviewer separately.

We attribute the issues with the ice module to the lack of snow parameterisation in GOTM (L. 158-161). Although onset of ice is not affected by this, the offset (as predicted by the ice module) is likely to occur too early due to the lack of insulation that the snow provides. This is unfortunate, and an ice module including snow (such as Simstrat's) would have been better, but the choice for GOTM enabled the use of the WET model and its elaborate description of biogeochemical processes. We used the temperature-threshold to compensate for this issue, although it is not optimal (as explained further below as well).

The discrepancies in stratification onset had indeed to do with the time windows and density thresholds. Although the model simulates bottom-top density difference accurately most of the years (as seen in Supplement figure S5), the observed data is noisier than the modelled data, and the threshold approach occasionally defined different periods as the onset. We tested multiple thresholds (both regarding the time window and density difference), but there were some consistent mismatches regardless of the choice of threshold value (despite such methods being well-established in literature). Still, our method is rather well-established in literature and these issues are unlikely to have an effect for the climate projections, as the degree of noise in the signal will be constant in the model. Also in reply to Reviewer 2's comment 2B, we clarified this further in the text (L. 215-216).

The shorter calibration period for the discharge was due to lack of measured discharge data before 2007. We do not understand the comment about Figure 2 in connection to the shorter calibration period for discharge, as the discharge did not seem to have a worse fit during the validation period. Regarding ice-off, we do indeed see a worse performance in 2020 and 2021. Rather than attributing this to the validation period, we expect this had to do with the exceptionally (though in coming decades perhaps normal) short period of ice cover in those years. The temperature-threshold approach seemed to perform less well in such years. We have added this information to the manuscript (L. 211-212, Table S2). Moreover, since years with short or no ice cover will become more frequent, we also added a line to the Discussion how this may impact our future projections (L. 270-273).

Although the shoreline is not too far away from the monitoring location, there are no major inflows anywhere near, as the largest part of the watershed is to the west of the lake. The measurement location is at the edge of the main basin of the lake, and for example seiche movements are occasionally visible in the high-frequency data (though they have a frequency around one day, so disappear with daily averaging). Regarding the processes under study here, we do not foresee a major effect of the location where the measurements were chosen, though we acknowledge that measurements from a single location are only moderately representative of the whole lake. Ice cover is likely longer in secluded bays compared to the main basin, stratification can form in shallow areas first (thermal bars), and blooms may occur in bays while the main basin is less affected, but overall, we expect our measurements to be moderately representative, due to the open connection to the main basin.

We appreciate the reviewer's comment regarding using the full period for calibration, with potential validation at the start. Although the reviewer did not request particular changes to be made to the manuscript, we will take the opportunity to elaborate on this issue, perhaps for no other reason than that we find it interesting as well! We

wanted to use the same model setup as Jimenez-Navarro et al. for reasons of clarity, but in general, this is an interesting proposition and it could be considered whether the model would have been more accurate if we had considered all data for calibration. In our view, the degree to which models have been established and are prone to overfitting, plays a large role here. Hydrological and hydrodynamical models are based on purely physical equations that have been widely applied (even if the models themselves have not) and are usually not heavily calibrated, so that overfitting, or other issues related to model stability, are less of a big risk. In biogeochemical models, however - at least the rather complex type that we used here -, many parameters are calibrated, many different equations are in use that describe the same process, overfitting is a real risk, and pools of N or P running dry can easily lead to unrealistic projections. A separate validation period may help to partially countermeasure these issues. Likewise, a validation period at the start could have downsides if there is still an effect of initial conditions, as biogeochemical models may need a longer spin-up than physical models. So, in this sense, we wonder if the recommendations of Shen et al. could/should be extended to biogeochemical models. Yet at the same time, data availability (both in frequency and period of coverage) is more pressing for biogeochemical variables, so being able to use the full period for calibration would have additional benefits as well. In short, we consider this topic outside the scope of our present study, but absolutely see the importance of looking further into this. We believe that the aquatic modelling community would benefit from an open discussion on this topic and indeed numerical testing of various methods, to find the advantages and limitations of different calibration and validation strategies.

3D Lake processes are heavily dependent on local atmospheric conditions, so to for the drainage area processes. The authors used five GCM models which by their global nature are course resolved. The GCMs are bias correction toward local measurements in Jiménez-Navarro et al. (2023), but if I understand Supplementary table E 4th column (RMSE) this bias correction is almost nonexistent.  Taking the difference between GCM INM-CM5-0 and measurements as an example, mean air temperature RMSE (Root Mean Square Error) drops from the unbiased comparison of 5.712 $^o$K to 5.687 $^o$K after bias correction and for Wind Speed from 4.283 to 2.588 m/s, and improvement with <1% and ~40 % respectively. The bias correction of precipitation, a key input to the drainage area model, looks to have failed. Now I might misunderstand how the Bias correction results are shown, but this illustrate my first point. Can we trust that the calibration is still valid using the climate models as input? Additionally, the reader needs to know why these climate models and scenarios were selected. I suspect because they cover the extreme ranges of for example temperature, precipitation, wind speed etc.. Furthermore since the setup is used for projecting climate effects, is the time frame (for drainage area and lake) long enough so that the models capture the climate trend (which is small compared to seasonal variations)? It would help the reader to see how the trends during the setup/calibration period are in the model compared to measurements.

We would argue that our bias-correction succeeded: the quantile mapping method that was used only aims to decrease the bias. The Supplementary table in Jiménez-Navarro et al. (2023), referred to by the reviewer, indeed shows that Bias significantly decreased and, in many cases, also RMSE, though there were cases in which RMSE slightly increased. With regard to precipitation, the bias correction did not fail, but we admit that we should have used scientific notation to show the RMSE and Bias values, which now appear to be 0 due to the unit (kg/m2/s, or mm/s). For example, for GCM INM-CM5-0, the precipitation bias before correction was $-2.7 \cdot 10^{-6}$ mm/s, and after bias

correction it was -8.5·10$^{-7}$ mm/s. In some combinations of GCM and variable, the bias correction had indeed little effect, but only because the GCM prediction already was relatively unbiased compared to observations.

A high NSE and low RMSE cannot be expected, as a hindcasted GCM is not intended to simulate the same weather events as were observed (e.g. a storm may pass at a different time than observed), but it should rather reflect the observed weather over a longer temporal scale (as opposed to a reanalysis dataset, which does intend to match observations as close as possible). A biased GCM, however, would be an indication of a biased prediction, and it is this that the quantile mapping mitigated. We therefore don't consider this study to be less accurate in terms of its future projections than other studies, though of course these projections present a large degree of uncertainty.

The five GCMs were selected because they represented a wider range of predictions compared to a single projection, but another main reason was that these GCMs provided all the necessary forcing needed to run both SWAT+ and GOTM-WET. Some other GCMs, for example EC-Earth-Veg and GFDL-ESM4 that were included in an earlier study using SWAT+ in Lake Erken (Jiménez-Navarro et al. 2021, doi: 10.3390/f12121803), missed some variables that were needed to run GOTM-WET (at least without using a different approach compared to the other GCMs).

We now state in the manuscript that these GCMs provided the required forcing and that they were bias-corrected (L. 135-137).

As in many climate projection studies, the time frame with measurements is indeed comparatively small to detect climatic trends, and the projected simulations are longer than the period with measurements itself. To test whether our model detected climatic trends during the calibration and validation period, we selected several model output variables that were predicted to show a trend with warming in Jiménez-Navarro et al. (2023): discharge, water temperature, oxygen, and NH$_4$ concentration. Both 3- and 15-m depths were assessed and annual averages were taken, and for simplicity, gaps in observations were linearly interpolated in order to fit a Mann-Kendall test. At a 0.05 significance level, according to the Mann-Kendall test, only corresponding trends in 3- and 15-m simulated oxygen concentration could be found, and in the observations, none of the variables showed a similar trend as in the climate simulations. In short, the reviewer's question could therefore be answered with "No, the calibration/validation time frame is not long enough to capture a climate trend". It should be noted that over longer periods, climatic trends in historical data of Erken do become visible for physical parameters at least (see Moras et al., 2019, cited in manuscript). Still, we do not consider this a restriction for this study. Considering biogeochemical data, Lake Erken has a comprehensive dataset, covering a longer period than most other sites (even longer data is available, but less regular and less variables, which is why we do not model even further back). As such, for this type of studies, it presents an optimal site to do this, and a lack of climatic trends over a comparatively short period in both observations and model, does not invalidate the use of the model itself. Since these findings are more in line with the "general" model performance, we did not see a convenient place in the manuscript to add this information, as this would rather be added to the manuscript by Jiménez-Navarro et al. (2023). In the current paper, readers can see both observed and simulated spring event timing over the period with observed data in Figure 2. Nevertheless, we hope that this information satisfies the reviewer.

**3E** Through the analysis of trends from the climate simulations, the authors treat the climate scenarios as constant change over time ex. Fig 3. This is not correct, in fact the gradient for each scenario change over time, especially for SSP 245. I suggest dividing the model output into 30 year chunks while conducting the analysis, or look at the amount of change from a reference to a far future period.

Although the air temperature change is indeed not linear (at least for SSP 2-45), we chose a linear model because it fitted the response well (Figure 3). We retained the use of a linear model for ease of communication and to facilitate a comparison with previous studies, which report phenological trends often as well as "x days per decade". From a practical point of view, the Mann-Kendall analysis allowed us to also assess relative changes. However, we agree that reporting the output as suggested by the reviewer has benefits, and it could further facilitate comparison with other studies and future meta-analyses. We now additionally report the values for the chunks 1985-2014, 2040-2069, and 2070-2099 in the Supplement section S4. It can indeed be seen here that some future trends do not seem to behave linearly, such as the chlorophyll peak date under SSP 2-45 (although it should be noted that the mid-century period is not in the middle of the other two periods). Additionally, we shortly discuss linearity and how the slopes should be interpreted, in the Discussion (L. 306-313).